



# Assessment of the total precipitable water from a sun photometer, microwave radiometer and radiosondes at a continental site in southeastern Europe

**Konstantinos Fragkos**[1], **Bogdan Antonescu**[1], **David M. Giles**[2,3], **Dragoş Ene**[1], **Mihai Boldeanu**[1], **Georgios A. Efstathiou**[4], **Livio Belegante**[1], and **Doina Nicolae**[1]

[1]National Institute of R&D for Optoelectronics INOE 2000, 409 Atomistilor Str., Măgurele, Ilfov, Romania
[2]Science Systems and Applications Inc. (SSAI), Lanham, MD 20706, USA
[3]NASA Goddard Space Flight Center (GSFC), Greenbelt, MD 20771, USA
[4]Department of Mathematics, Centre for Geophysical and Astrophysical Fluid Dynamics, University of Exeter, Exeter, UK

**Correspondence:** Konstantinos Fragkos (kostas.fragkos@inoe.ro)

**Abstract.** In this study, we discuss the differences in the total precipitable water (TPW), retrieved from a Cimel sun photometer operating at a continental site in southeast Europe, between version 3 (V3) and version 2 (V2) of the AErosol RObotic NETwork (AERONET) algorithms. In addition, we evaluate the performance of the two algorithms comparing their product with the TPW obtained from a collocated microwave radiometer and nearby radiosondes during the period 2007–2017. The TPW from all three instruments was highly correlated, showing the same annual cycle, with lower values during winter and higher values during summer. The sun photometer and the microwave radiometer depict the same daily cycle, with some discrepancies during early morning and late afternoon due to the effect of solar zenith angle on the measurements of the photometer. The TPW from V3 of the AERONET algorithm has small differences compared with V2, mostly related to the use of the new laboratory-based temperature coefficients used in V3. The microwave radiometer measurements are in good agreement with those obtained by the radiosonde, especially during night-time when the differences between the two instruments are almost negligible. The comparison of the sun photometer data with high-quality independent measurements from radiosondes and the radiometer shows that the absolute differences between V3 and the other two datasets are slightly higher compared with V2. However, V3 has a lower dependence from the TPW and the internal sensor temperature, indicating a better performance of the retrieving algorithm. The calculated one-sigma uncertainty for V3 as estimated, from the comparison with the radiosondes, is about 10 %, which is in accordance with previous studies for the estimation of uncertainty for V2. This uncertainty is further reduced to about 6 % when AERONET V3 is compared with the collocated microwave radiometer. To our knowledge, this is the first in-depth analysis of the V3 TPW, and although the findings presented here are for a specific site, we believe that they are representative of other mid-latitude continental stations.

## 1 Introduction

Water vapour is a crucial atmospheric component of Earth's climate since it is the most abundant greenhouse gas (IPCC, 2013). Water vapour plays a prominent role in the hydrological cycle through water evaporation and condensation while providing the energy to drive moist convection and resulting precipitation. The large-scale flow and local circulations contribute to the large variability of the spatial and temporal distribution of water vapour. For weather forecasting, precipitation efficiency is strongly related to the water vapour content, which in turn determines the potential stability of the atmo-

spheric column. Thus, accurate estimations of water vapour content are essential for meteorological and climate applications such as radiative transfer modelling (e.g. Paynter and Ramaswamy, 2012) or weather forecasting (e.g. Liang et al., 2015).

A common measure of the water vapour content in the atmosphere is the total precipitable water (TPW), defined as the total water contained in a column of unit cross section extending all the way from the earth's surface to the top of the atmosphere (American Meteorological Society, 2018). Initially, radiosonde measurements were used to measure TPW (e.g. Reber and Swope, 1972). Although, the radiosonde measurements are reliable they are limited, for example, by freezing of moisture sensors, which leads to errors in the estimation of moisture, or by the phase lag between the dry and wet bulb sensors (Campmany et al., 2010). In addition, the global radiosonde network coverage is limited (e.g. McCarthy, 2008). Thus, considering the large variability of water vapour both in time and space, it becomes obvious that soundings provide a very limited spatio-temporal representation of TPW (Liang et al., 2015).

To overcome these issues, a number of methods for TPW estimation based on active or passive remote-sensing techniques, either from the ground or the space, have been developed. From the ground the most common ones include the GPS system (Mears et al., 2015), microwave radiometers (Westwater and Guiraud, 1980), Cimel sun photometers (Halthore et al., 1997; Holben et al., 1998), Fourier transform infrared spectroscopy (Sussmann et al., 2009) and Raman lidars (Ferrare et al., 1995; Filioglou et al., 2017). Recently, techniques have been developed for the retrieval of TPW from measurements of the precision solar spectroradiometer at the World Radiation Center (WRC) Davos (Raptis et al., 2018), the PESR/PREDE-POM sun–sky radiometers (Campanelli et al., 2018) and from MAX-DOAS observations (Wagner et al., 2013).

The quality of the retrieved TPW from each instrument is assessed through comparison with other independent measurements. In general, radiosondes and the global GPS systems have been used for the evaluation of TPW measurements from satellite data (Van Malderen et al., 2014; Román et al., 2015; Vaquero-Martínez et al., 2017a, b, 2018; Gui et al., 2017). Of particular interest is the evaluation of the TPW from the Cimel sun photometer that is part of the AErosol RObotic NETwork (AERONET), a network with global coverage. Several studies have validated the TPW retrieval from Cimel sun photometer with radiosondes, GPS and microwave radiometer measurements (e.g. Sapucci et al., 2007; Schneider et al., 2010; Campmany et al., 2010; Pérez-Ramírez et al., 2014; Van Malderen et al., 2014; Gui et al., 2017; Campanelli et al., 2018). Although their network is dense, the Cimel sun photometer has a series of limitations because they require sunlight, which indicates that at least the solar disc must be free from clouds for TPW retrieval. These conditions restrict the availability of data just during daytime

and thus reduce the temporal availability of the datasets. Nevertheless, Pérez-Ramírez et al. (2014) demonstrated that the Cimel sun photometer can provide extended time series with good temporal resolution. A lunar photometer could provide TPW during night-time (e.g. Barreto et al., 2013), but this product is not yet available in the AERONET database.

In this article, we focus on measurements conducted at the Romanian Atmospheric 3D research Observatory (RADO). The reason for this is that RADO is the only site, to our knowledge, in southeastern Europe that has long-term measurements of TPW from three independent instruments: Cimel sun photometer, microwave radiometer and radiosondes. Therefore, it can be used as a test bed to assess the quality of the measurements, especially because the radiometer provides continuous high-quality observations of TPW. Furthermore, this site is one of the few potential sites from southeastern Europe that can be used for satellite calibration and validation activities. Thus, the evaluation of the RADO measurements is an essential process towards this goal. Recently the newly released version 3 of the AERONET products has become publicly available. This new version incorporates significant improvements for direct sun measurements, such as a new, improved cloud screening algorithm, automated quality check procedures, inclusion of higher air mass data, and new temperature characterisation and corrections to all channels (Giles et al., 2019). To our knowledge, no study has evaluated the newly released version of the TPW from the AERONET. In this study, the quality of the TPW measured by three different instruments (i.e. HATPRO-G2 microwave radiometer, Cimel sun photometer and Vaisala RS92 radiosondes) at a site in southeastern Europe is assessed. The paper is organised as follows. The instruments used in this study are described in Sect. 2. In Sect. 3, the climatology of the annual cycle of TPW observed over the study area and the comparison of the different datasets employed for the measurements of TPW are presented. More specifically, the differences between the microwave radiometer and radiosondes, and Cimel V2 and V3 and the radiosondes, Cimel V2 and V3 and the radiometer are analysed, and the factors affecting their agreement are assessed. Section 4 summarises this article.

## 2 Data and methodology

### 2.1 Meteorological parameters

The HATPRO-G2 microwave radiometer and the Cimel sun photometer used in this study were located at the Romanian Atmospheric 3D Observatory (RADO, 44.82° N, 26.82° E, 93 m a.s.l.), part of the National Institute of Research and Development for Optoelectronics (INOE2000). The observatory is located in the city of Măgurele, Ilfov, at the central part of the Romanian plain, approximately 10 km southwest of Bucharest, the capital city of Romania, and is surrounded

by research facilities, residence buildings and a small forest. The central Romanian plain has a temperate climate influenced by the western circulation, the east European anticyclone, the Mediterranean cyclones and the tropical advection (Cheval et al., 2009). The relative humidity at Măgurele, as calculated from observations from the RADO weather station between 2007 and 2016, has high values (> 80 %) during November–February and low values (< 60 %) between May and September (Fig. 1). Since the Cimel sun photometer performs measurements only when the solar disk is free of clouds, two critical parameters for the availability of the Cimel sun photometer data are the sunshine duration and the cloud fraction. For the calculation of the climatology of the sunshine duration the Surface Radiation Data Set – Heliosat (SARAH) – Edition 2 (Pfeifroth et al., 2017) of the EUMETSAT's Satellite Application Facility on Climate Monitoring (CM SAF) was used. The sunshine duration (SDU) product is the daily sunshine duration per day at which direct normal irradiance (DNI) exceeds the WMO threshold of 120 W m$^{-2}$. SDU is derived by the ratio of sunny slots to all slots during daylight multiplied by the length of day. The length of day is calculated depending on the date, longitude and latitude. The length of day is restricted by a threshold of the solar elevation angle (SEA) of 2.5°. The SDU product is provided on a regular latitude–longitude grid with a spatial resolution of 0.05° × 0.05°. In this study, for the calculation of the climatological sunshine duration, the daily SDU at the closest pixel over Măgurele during the period 2005–2015 was used. A full description of the SDU product can be found at Kothe et al. (2017). For the calculation of the cloud fraction climatology, the CM SAF cloud property dataset using SEVIRI – edition 2 (CLAAS-2; Finkensieper et al., 2016) was used. The cloud fractional cover (CFC) is defined as the fraction of cloudy pixels per grid cell compared to the total number of analysed pixels in the grid cell and is expressed as a percentage. In this study the daytime CFC during the period 2005–2015 was used. The daily CFC product is provided on a regular latitude–longitude grid with a spatial resolution of 0.05° × 0.05°. A full description and evaluation of the CFC product is given in Benas et al. (2017). High cloud coverage (> 70 %) affects the RADO site from November to February (Fig. 1), while the lowest cloud fraction (< 40 %) is during July–August. The rest of the months the cloud fraction ranges between 50 % and 60 %. The high percentage of clouds, in combination with the small sunshine duration (Fig. 1), during late autumn and winter affects the availability of the Cimel sun photometer data during these months. Thus calculations of multi-year annual mean TPW values from Cimel sun photometer observations are biased from the highest number of data points during summer. The sunshine duration exhibits a clearly annual cycle with a minimum during winter and maximum during summer and ranges from ∼ 2.3 h during January (minimum) to up to more than 10 h during July (maximum) (Fig. 1).

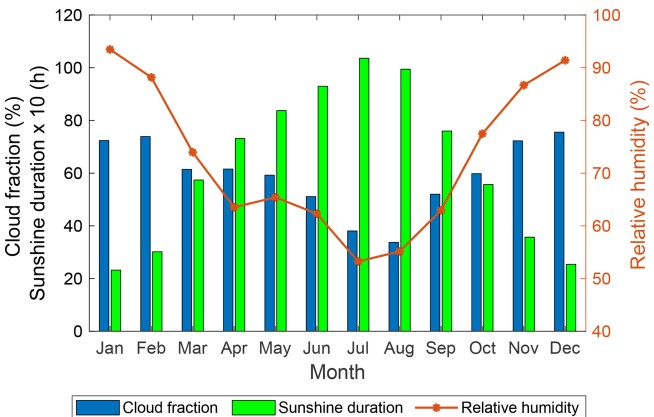

**Figure 1.** Annual cycle of the cloud fraction, sunshine duration and relative humidity at Măgurele (adapted from Carstea et al., 2019 Fig. 2).

## 2.2 Cimel sun photometer

A Cimel Electronique 318A sun photometer (serial number 359) was installed at the RADO facilities in July 2007 and was operated until May 2016, when it was reallocated to Poland. As a replacement a Cimel lunar photometer has been operating since 2016, but data from this instrument have not been used in this study due to the limited availability of level 2 data. The Cimel sun photometer is the standard instrument of AERONET (Holben et al., 1998) used for the study of the aerosol total column load. It performs spectral measurements of the direct sun irradiance and sky radiance at six discrete wavelengths using interference filters. The filters are centred at the wavelengths of 340, 380, 440, 500, 675, 870 and 1020 nm. An additional channel at 935 nm is used for the retrieval of the TPW. The instrument is calibrated almost annually following the procedures and the guidelines of AERONET. TPW is calculated based on a modified expression of the Beer–Bouguer–Lambert law. Since Giles et al. (2019) provide a full description of the TPW retrieval algorithm (see Sect. 2 of that paper), in this section just the major differences between V2 and V3 and some other factors that may influence the TPW retrieval are discussed. For the computation of TPW a necessary preliminary step is the subtraction of the AOD and Rayleigh optical depths from the total optical depth at 935 nm. Since AOD is not calculated directly for the 935 nm channel due to the strong effect of water vapour, the AOD at 870 nm is extrapolated at the 935 nm using the Ångström exponent (AE) at 440–870 nm. The main differences in the computation of TPW in V3 are that the new algorithm accounts for an updated continuum look-up table (Mlawer et al., 2012), using total internal partition sums (Gamache et al., 2017) and using the extraterrestrial spectral solar irradiance from Coddington et al. (2016). In this study all available data from July 2007 to May 2016 for level 2 from versions 2 and 3 of AERONET algorithms

were used. Level 2 data are screened for clouds, quality controlled, and pre-field and post-field calibrations are applied. The newest released version 3 incorporates improvements for the direct sun measurements (1) related to the screening of clouds, (2) the automated data quality assurance, (3) inclusion of data with higher air masses (i.e. from 1 to 7, in contrast with V2 that ranges from 1 to 5) and (4) implementation of spectral temperature corrections based on laboratory measurements (i.e. unlike version 2 that was based on the manufacture specifications). Details about all the improvements implemented in V3 of AERONET can be found at Giles et al. (2019). The AERONET TPW measurement uncertainty is estimated to be < 10 % (Halthore et al., 1997; Holben et al., 2001), which is consistent with the one-sigma uncertainty for AERONET V2 provided by Pérez-Ramírez et al. (2014) of 7 %–9 % after evaluating the TPW from AERONET at the U.S. Department of Energy Atmospheric Radiation Measurement Program (ARM) sites against microwave radiometers, GPS and radiosondes.

## 2.3 Microwave radiometer

The HATPRO-G2 microwave radiometer used in this study was produced by Radiometer Physics GmbH. It is a passive instrument working in the microwave regime. It consists of two working bands at 22–31 and 51–58 GHz, each with seven channels. The relevant receiving optics, the ambient load, the internal scanning mechanism, the electronics and the data acquisition system of the radiometer are described in Rose et al. (2005). For humidity profiling only the first band is used. The vertical resolution for profiling is variable, ranging from 200 below 2000 to 800 m for altitudes higher than 5000 m. Water vapour emission dominates the signal in the 23.8 GHz channel, which is on the wing of the 22.2 GHz water vapour absorption line, whereas liquid water emission constitutes the primary portion of the signal at 31.4 GHz (Turner et al., 2007). From these two observations, both integrated water vapour (IWV) and liquid water path (LWP) can be retrieved. The retrievals are performed in the zenith direction. In this study, the IWV was used, which presents, according to the manufacturer (RPG-HATPRO-G4 series microwave radiometers for continuous atmospheric profiling, available at https://www.radiometer-physics.de/download/PDF/Radiometers/HATPRO/RPG_MWR_PRO_TN.pdf, last access: 13 July 2018), an accuracy of $\pm 0.2$ kg m$^{-2}$ RMS and noise of 0.05 kg m$^{-2}$. Considering the density of liquid water, the IWV expressed in kilograms per square metre is equivalent with the TPW expressed in millimetres of liquid water (Bevis et al., 1992). In this study measurements are performed each 2 s. To ensure the high quality of measurements, the instrument is absolutely calibrated with liquid nitrogen every 6 months following the instructions of the manufacturer. All the available data between 16 December 2009 and 31 December 2017 were used. The internal data quality has three options for filtering level 2 data (retrieved atmospheric data). The "Flag Data Quality" (level 2) option does not filter the level 2 data according to the quality level but flags each data sample in the rain flag byte. With the option "Remove Medium/Low Q.", medium- and low-quality samples are not transmitted by the radiometer. In this case, the sample sent to the personal computer that controls the instrument is the repeated latest high-quality sample. The filter "Remove Low Quality" only removes the worst-quality data and transmits high- and medium-quality data. In the present study, the first option was used for the creation of the level 2 data; thus only data that have been flagged as rain from the internal sensor of the instrument have been removed. In addition, all days with data have been visually inspected for identification of instrumental malfunctions, which can include periods when there are no changes in the TPW values due to bad transmission of data or periods with low-quality data (i.e. when the TPW remained high after rain, until returning to its previous levels after some time).

## 2.4 Radiosondes

The radiosonde measurements were obtained from the sounding database maintained by the University of Wyoming (http://weather.uwyo.edu/upperair/sounding.html, last access: 13 July 2018). Between July 2007 and December 2017, 3760 radiosonde measurements for 00:00 UTC and 3759 for 12:00 UTC were available from the Bucharest site, situated at approximately 30 km northeast from the RADO facility and operated by the Romanian National Meteorological Administration. The radiosondes used during the study period were of the Vaisala RS92 type. For this type of radiosondes, Miloshevich et al. (2009) showed that the accuracy of the humidity sensor during daytime depends on the calibration error and the dry bias due to the solar heating effect (Turner et al., 2003) and during the night-time just from the calibration error. The overall uncertainty of TPW from radiosonde measurements has been estimated to be $\pm 5$ % (Pérez-Ramírez et al., 2014). TPW over the entire sounding was calculated as

$$\text{TPW} = \frac{1}{\rho g} \int_{p_1}^{p_2} x \, dp, \tag{1}$$

where $x(p)$ is the water vapour mixing ratio at the pressure level $p$, $\rho$ is the density of water and $g$ is the acceleration of gravity.

## 2.5 Methodology

For the computation of the daily mean values of TPW all available measurements that qualify the quality criteria were used. A preliminary step was the averaging of TPW from the microwave (MWV) radiometer into 1 min intervals. Table 1 gives an overview of the total number of observations, along with their total number of corresponding days that have been analysed for each instrument and for the different versions

of the AERONET algorithms in order to compute the daily averages. Due to the different schedule of each instrument and the gaps in each database, the computed averages cannot be directly compared between them. For a direct comparison we extracted the common measurements between V2 and V3 and they were averaged for ±20 min around the launch time of the noon radiosonde. The same averaging was applied to the MWV radiometer data, so as to extract a dataset of simultaneous or nearly simultaneous measurements from all instruments. Since the exact hour of the radiosonde launch is not explicitly known, this 40 min interval has been selected in order to ensure that the instruments detect the same air masses and to limit the atmospheric variability that takes place on timescales larger than 1 h (Schneider et al., 2010). If the GPS information of the radiosondes is available, a further improvement in the coincidence criteria would be to average the Cimel sun photometer data for ±20 min since the time the balloon reaches the altitude of the 4 km, following Schneider et al. (2010). However, in our case access to the raw data is not available; thus the averaging was performed ±20 min around the launch time. For the comparison of the MWV radiometer and Cimel sun photometer data with the radiosondes, the same coincidence criteria as described above were used. The comparison of the two different algorithms of AERONET is based just on their common measurements. This way the comparison provides insight into the TPW calculation differences between the two algorithm versions rather than impacts due to cloud screening and instrument quality controls. For the comparison of the Cimel sun photometer data with the MWV radiometer, the exact time matched measurements were selected. For the evaluation of the MWV radiometer and Cimel sun photometer data, the radiosonde TPW was used as the reference measurements because they are considered more representative of the actual atmospheric conditions. However, since the radiosonde site is at a distance of ∼ 30 km from the RADO facilities, there is the possibility that the different instruments detect air masses with different characteristics, especially when the radiosondes are affected by southwest winds. Thus, the calculated uncertainty expressed as the $1\sigma$ of the mean difference among the different datasets is expected to be little overestimated when compared to the radiosondes. The absolute and relative differences between two sets of measurements were defined as

$$X - X_{\mathrm{ref}} \tag{2}$$

and

$$100 \cdot \frac{(X - X_{\mathrm{ref}})}{X_{\mathrm{ref}}}, \tag{3}$$

respectively, where ($X_{\mathrm{ref}}$) is the reference measurement (i.e. the radiosonde measurement, except for the comparison between the Cimel sun photometer and the microwave radiometer).

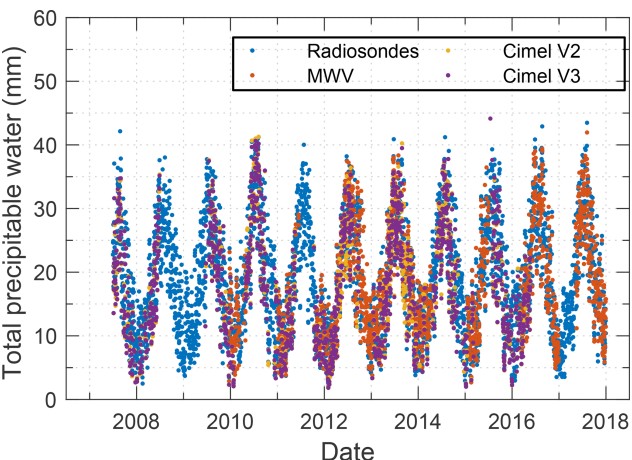

**Figure 2.** Time series of the daily mean values of the total precipitable water during the period 2007–2017 based on measurements from radiosondes (blue dots), a microwave radiometer (orange dots), and Cimel sun photometer version 2 (yellow dots) and version 3 (magenta dots) of the algorithm.

## 3 Results

### 3.1 Climatology of total precipitable water in Măgurele

The times series of the daily mean values for the TPW from the different instruments employed in this study are shown in Fig. 2. In general, the radiosonde measurements are available twice per day (i.e. 00:00 and 12:00 UTC). The Cimel sun photometer measurements are restricted only during daytime and under conditions that require the solar disc to be clear of clouds, while the microwave radiometer performs measurements during daytime and night-time under all weather conditions. Although there are differences in the measurement schedule, all three instruments depict the same annual cycle, demonstrating their capability of performing long-term measurements for climatological applications (Fig. 2). The gaps in Cimel sun photometer time series are due to the calibration of the instrument, which requires the reallocation of the instrument. Data gaps of the microwave radiometer are due to malfunction of the instrument or controlling personal computer (usually solved with a restart after a maximum of couple of days) or due to the relocation of the instrument during different measurement campaigns (data not included in this study). Furthermore, in the beginning of 2016 the instrument was sent to the manufacturer for testing and replacement of several components.

The observed differences in the mean values calculated from all instruments (Table 2) can be mostly attributed to the different operating period of each instrument and their different sampling rates. However, even though the overall mean from Cimel sun photometer measurements is not significantly different from the radiosondes and the microwave radiometer estimates, Cimel sun photometer measurements

**Table 1.** Overview of the measurement characteristics and datasets used in this study for the period 2007–2017.

| Instrument | Retrieval method | Total number of observations | Total number of daily mean values | Data frequency |
|---|---|---|---|---|
| Radiosondes | Thin-film capacitance relative humidity sensor use of balloons for vertical profiles | 7503 | 3784 | 12 h |
| Radiometer | Sky brightness temperature at 23.8 GHz water vapour absorption band | 1 859 315 | 1612 | 2 s |
| Cimel V2 | Solar direct irradiance at 940 nm absorption band | 33 324 | 1293 | $\sim$ 20 min for clear sky conditions |
| Cimel V3 | Solar direct irradiance at 940 nm absorption band | 35 373 | 1325 | $\sim$ 20 min for clear sky conditions |

**Table 2.** Summary of the daily mean statistics of all instruments and algorithms for the period from July 2007 to December 2017.

| | Radiosondes | MWV radiometer | Cimel V2 | Cimel V3 |
|---|---|---|---|---|
| Average (mm) | 18.75 | 17.47 | 18.86 | 18.58 |
| Standard deviation (mm) | 8.78 | 8.50 | 8.87 | 8.99 |
| Maximum (mm)/(date) | 43.48/(07.08.2017) | 41.95/(07.08.2017) | 41.27/(08.08.2010) | 40.22/(08.08.2010) |
| Minimum (mm)/(date) | 1.87/(01.02.2012) | 2.04/(01.02.2012) | 1.83/(01.02.2012) | 1.83/(01.02.2012) |

are actually biased towards the higher TPW values observed during the summer. Since the cloud fraction during the winter months at Măgurele is pretty high, more than 70 % from November to January (Fig. 1) when TPW also attains its minimum values (Fig. 2), the number of Cimel sun photometer observations is substantially reduced, leading to the inclusion of a reduced number of low-TPW days in the Cimel sun photometer dataset. This observed summer (wet) bias is partly compensated for by the inherent Cimel sun photometer dry bias (e.g. Schneider et al., 2010) due to restrictions of measurements when the solar disc is cloud free and thus the overall TPW mean from the Cimel sun photometer is similar to the other methods (Table 2). This dry bias for the mid-latitudes is more pronounced during winter and can range from 25 % to 50 %, while in summer it ranges from 5 % to 25 % (Gaffen and Elliott, 1993). The clear-sky monthly bias can be clearly seen in the mean monthly values of TPW (Table 3), for which the Cimel sun photometer measurements during January can be lower by $\sim$ 25 % compared to the radiosondes while the summer mean monthly values are lower by only a few percent (e.g. $\sim$ 4 % for August) (Fig. 4a). Such behaviour is not observed for the MWV radiometer, with the differences in their mean monthly values ranging within $\pm 10$ % for all months (Fig. 4b). The minimum values daily values can be as low as 2 mm, while the maximum values exceed 44 mm (Table 2). The peak-to-peak range during the year (i.e. from minimum to maximum) can be up to 20 mm.

The annual cycle of the TPW as depicted by all three instruments has a minimum during winter months (DJF) and a maximum during summer months (JJA) (Fig. 3). Higher air temperature during the summer implies a larger capacity to store water vapour without saturation (Campmany et al., 2010). The small differences in the monthly median values for all instruments are due to their different sampling rates. For example, the increased number of outliers in the radiosonde box plots, compared to the other instruments, can be attributed to the limited number of measurements (i.e. a maximum of two per day). Thus, some high or low values are not smoothed by averaging all measurements during the day (Fig. 3a). In any case, the main aim of the analysis presented here is to show that the annual cycle of TPW can be depicted fairly well by all instruments and demonstrate their capabilities for long-term monitoring for climatological applications. A direct comparison of the daily values from each instrument is not valid due to the very different sampling rates and the diurnal variation in TPW as shown in Fig. 5. An overview of the statistical values based on all available measurements for all three instruments is shown in Table 3. A dataset was constructed, as described in Sect. 2.4, containing the common measurements and thus allowing for a direct comparison among the three instruments. This dataset consists of a total of 234 days during the measurement period, which is limited by the Cimel sun photometer observations during conditions in which the solar disc was free of clouds. For this reason, the comparison of the different instruments is not affected by the clear-sky dry bias. An overview of the long-term averages of the common measurements from all instruments can be seen in Table 4. The MWV radiometer has the higher mean TPW (18.57 mm), followed by the radiosondes (17.96 mm), Cimel V2 (17.80 mm) and finally Cimel V3 (17.65 mm). Although

**Table 3.** Mean monthly and median values of TPW and their IQR from the different instruments used in this study. All units are in millimetres.

| Month | Radiosondes | | | Radiometer | | | Cimel V2 | | | Cimel V3 | | |
|---|---|---|---|---|---|---|---|---|---|---|---|---|
| | Mean | Median | IQR | Mean | Median | IQR | Mean | Median | IQR | Mean | Median | IQR |
| January | 9.69 | 9.12 | 5.65 | 10.40 | 10.39 | 5.83 | 7.27 | 6.62 | 4.99 | 7.66 | 6.98 | 5.53 |
| February | 10.09 | 10.14 | 6.28 | 10.20 | 10.12 | 5.98 | 9.04 | 9.10 | 6.79 | 8.99 | 8.35 | 6.03 |
| March | 11.42 | 11.04 | 5.94 | 11.49 | 10.87 | 5.71 | 10.05 | 9.40 | 5.37 | 9.81 | 9.17 | 5.00 |
| April | 15.17 | 15.18 | 6.27 | 15.54 | 15.74 | 6.03 | 14.03 | 14.20 | 4.50 | 13.94 | 14.00 | 4.90 |
| May | 20.92 | 20.69 | 7.63 | 21.73 | 21.98 | 7.43 | 19.69 | 19.41 | 7.33 | 19.36 | 19.16 | 6.43 |
| June | 27.23 | 27.58 | 7.47 | 28.58 | 28.70 | 6.64 | 26.47 | 26.84 | 7.89 | 26.36 | 26.91 | 7.94 |
| July | 29.64 | 29.67 | 8.11 | 29.78 | 30.45 | 6.88 | 27.82 | 27.91 | 8.32 | 28.28 | 28.20 | 8.05 |
| August | 28.95 | 29.44 | 8.25 | 28.11 | 29.05 | 8.60 | 27.75 | 27.77 | 7.50 | 27.71 | 27.77 | 7.64 |
| September | 23.32 | 22.87 | 9.55 | 22.36 | 22.06 | 7.43 | 20.71 | 20.35 | 7.05 | 21.41 | 20.86 | 7.37 |
| October | 18.30 | 18.38 | 9.59 | 18.79 | 17.86 | 8.24 | 15.41 | 14.47 | 9.06 | 15.28 | 14.49 | 8.25 |
| November | 14.37 | 13.95 | 7.47 | 14.15 | 13.71 | 6.39 | 12.00 | 11.13 | 6.27 | 11.65 | 10.75 | 6.13 |
| December | 10.62 | 9.83 | 6.41 | 10.65 | 10.06 | 5.54 | 9.23 | 9.00 | 5.88 | 9.49 | 9.14 | 6.04 |

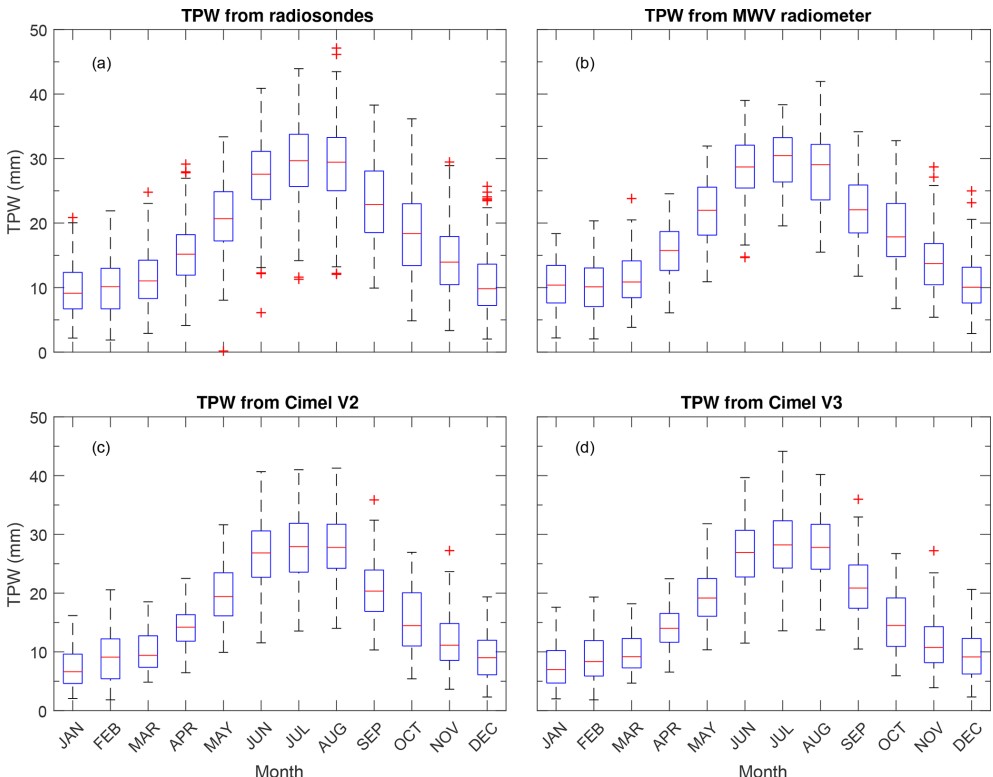

**Figure 3.** Monthly variation in total precipitable water from **(a)** radiosondes during the period 2007–2017, **(b)** microwave radiometer during the period 2009–2017, **(c)** Cimel sun photometer version 2 data and **(d)** Cimel sun photometer version 3 data for the period 2007–2016. The median values are shown as the red lines, the interquartile range (IQR) is spanned by the vertical bars and the whiskers show the 1.5 IQR. The red + symbols show the outliers in the datasets.

this dataset consists of nearly time-matched measurements, the small differences in the long-term averages may occur from differences in the geometry of the measurements and subsequently the sounding of air masses with different characteristics. For example, the Cimel sun photometer measures the direct sunlight and can track the sun between clouds, while the MWV radiometer measures the zenith sky radiance and it may not be completely cloud free for the same sky. The radiosondes are also launched from a different area, which could possibly track different air masses. The mean monthly TPW values (Table 5) appear to have very good agreement (within ±5 %) among the different instruments when using

common data periods (Fig. 4b). The Cimel sun photometer clear-sky dry bias that had been observed, especially during the winter months in the long-term averages when computed from all measurements (Table 3), has been cancelled out, as
can be clearly seen in Fig. 4b.

## 3.2 Sensitivity of the instruments to diurnal variation

As mentioned previously, the temporal resolution of the microwave radiometer on the order of a few seconds in combination with its capacity to operate under all weather con-
10 ditions allow the detection of the TPW diurnal variations. In addition, under clear-sky conditions, the Cimel sun photometer performs measurements at about every 15 min. To verify if both instruments depict the same daily cycle, the diurnal variability for 6 selected days was examined. The days were
15 selected to meet the following conditions: the Cimel sun photometer measurements cover most of the day and in particular for high solar zenith angles (SZAs, $> 70°$), no discontinuation due to clouds in Cimel sun photometer measurements was observed from sunrise to sunset and the measure-
20 ments cover all seasons. The Cimel instruments and MWV radiometer depict the same diurnal variation during daytime (Fig. 5), with some small differences in their absolute values that are further investigated in the following sections. For some of the selected days (i.e. 8 June 2012, 26 Octo-
25 ber 2013) there are differences in the diurnal variation during the early morning or late afternoon hours, which are most likely artefacts associated with direct sun measurements at high air masses (e.g. SZA $> 70°$). These artefacts are due to Cimel clock deviations that result in some minor deviation in
the optical air mass calculation and thus slightly impact AOD but within uncertainty expectations (see Sect. 3.3.1 of Giles et al., 2019).

## 3.3 Comparison between radiosondes and microwave radiometer

To account for spatial and temporal differences between the radiosonde and the microwave radiometer, all the microwave radiometer data were averaged over an interval of 40 min centred on the radiosonde launching time. A total number of 2820 common measurements, out of which 1416 during
daytime (i.e. at 12:00 UTC) and 1404 during night-time (i.e. at 00:00 UTC), were extracted for the comparison. The relative difference between the two datasets is in general within $±25 \%$ (Fig. 6). The MWV radiometer slightly overestimates TPW with the overall difference from the radiosondes to be
$1.82 ± 9.61 \%$ ($0.17 ± 1.66$ mm). This overestimation is more evident during daytime (i.e. $3.12 ± 9.93 \%$ or $0.35 ± 1.71$ mm) due to the radiation dry bias effect that affects the radiosondes (e.g. Vömel et al., 2007), which is more pronounced for TPW values less than 10 mm (Fig. 8b). During night-time
the differences are almost negligible (i.e. $-0.50 ± 9.10 \%$ or $-0.01 ± 1.57$ mm).

The two datasets are highly correlated (Fig. 7a; $R^2 = 0.97$), with the majority of the points over the $y = x$ line. However, for the higher values of TPW (i.e. TPW $> 30$ mm) an increased scatter of the data is observed, without be-
55 ing significantly high. The histogram of the relative differences between the two instruments, which peaks at about 1 % (Fig. 7b), does not follow a normal distribution, as indicated by the Shapiro–Wilk test for normality (Shapiro and Wilk, 1965) ($p$ value $< 2.2e–16$). About 96 % of the data are
60 within $±20 \%$, while $\sim 78 \%$ lie in the range of $±10 \%$. The difference between the two datasets has a small dependence from the TPW amount of $-0.169 \%$ mm$^{-1}$ (Fig. 8a). This dependence is more evident for the daytime measurements (i.e. for radiosondes launched at 12:00 UTC; Fig. 8b), while
for the night-time measurements the dependence is almost negligible (i.e. $-0.092 ± 0.052 \%$ mm$^{-1}$; Fig. 8c). The best agreement between the two datasets is achieved for TPW values ranging between 15 and 35 mm. The increased difference for TPW values higher than 40 mm cannot be fully evaluated
due to the very small number of observations (i.e. just 19 measurements).

## 3.4 Comparison of Cimel V2 and V3

To assess the differences of the TPW derived from the newly released version 3 from AERONET and the previous ver-
75 sion 2, only their common measurements were used. The difference in the number of observations between the two versions (see Table 1) arises from the fact that they have different quality control and cloud screening procedures (Giles et al., 2019). A total of 27 707 common observations be-
80 tween the two versions were extracted for comparison. In general, the differences between the two versions are small, ranging within $±2 \%$ and rarely exceeding 5 %, with V2 having higher values than V3 (Fig. 9a and b). The overall difference between the two datasets for the period 2007–2016
is $0.60 ± 20.91 \%$ ($0.08 ± 20.14$ mm). The differences at the AOD at 870 nm between the two different algorithm versions (Fig. 9c) are generally pretty low and rarely exceed the $±0.01$ AOD units. The cyclic nature of the AOD differences (Fig. 9c) suggests the variation in the AOD with tempera-
ture for version 2. The V2 data are not temperature-corrected for the 870 nm filter and this produces a difference in AOD between temperature-corrected (V3) and not-corrected (V2) data due to this specific filter before 2009. The 870 nm filter was changed in 2009 in this specific instrument and its
dependence on temperature was a magnitude lower than the initial filter. As a result, the filter used in the instrument from 2009 and onward shows less deviation from V2 since the temperature correction needed for the filter is minimal. This is a clear example of how implementation of temperature
correction in version 3 significantly improved the AOD and TPW, before 2009.

To further evaluate the differences between TPW from the two different versions of the AERONET algorithm, a series

**Table 4.** Same as Table 2 but just for the common measurements from all instruments.

|  | Radiosondes | MWV radiometer | Cimel V2 | Cimel V3 |
|---|---|---|---|---|
| Average (mm) | 17.96 | 18.57 | 17.80 | 17.65 |
| Standard deviation (mm) | 8.95 | 9.25 | 8.72 | 8.71 |
| Maximum (mm)/(date) | 39.90/(25.06.2013) | 38.31/(08.07.2012) | 36.35/(25.06.2013) | 36.02/(25.06.2013) |
| Minimum (mm)/(date) | 2.02/(25.01.2010) | 1.784/(25.01.2010) | 1.97/(25.01.2010) | 1.95/(25.01.2010) |

**Table 5.** Same as Table 3 but just for the common measurements. All units are in millimetres.

| Month | Radiosondes | | | Radiometer | | | Cimel V2 | | | Cimel V3 | | |
|---|---|---|---|---|---|---|---|---|---|---|---|---|
| | Mean | Median | IQR | Mean | Median | IQR | Mean | Median | IQR | Mean | Median | IQR |
| January | 6.41 | 4.95 | 6.13 | 6.67 | 4.87 | 6.20 | 6.53 | 4.76 | 6.38 | 6.44 | 4.67 | 6.31 |
| February | 8.21 | 6.99 | 6.88 | 8.86 | 7.57 | 6.01 | 8.67 | 7.41 | 5.98 | 8.51 | 7.26 | 5.92 |
| March | 9.17 | 8.06 | 3.95 | 9.43 | 8.78 | 3.32 | 9.21 | 8.93 | 3.25 | 9.08 | 8.73 | 3.18 |
| April | 14.48 | 14.37 | 6.79 | 15.11 | 15.07 | 5.22 | 14.30 | 14.37 | 4.59 | 14.16 | 14.14 | 4.76 |
| May | 19.14 | 18.46 | 7.75 | 19.09 | 19.06 | 7.69 | 18.42 | 18.25 | 7.33 | 18.27 | 18.25 | 8.49 |
| June | 27.51 | 27.50 | 7.69 | 28.46 | 28.42 | 7.96 | 27.32 | 27.34 | 7.89 | 26.36 | 26.91 | 8.18 |
| July | 28.21 | 28.61 | 6.81 | 29.49 | 30.18 | 8.12 | 27.93 | 29.21 | 8.18 | 27.12 | 27.02 | 6.30 |
| August | 29.42 | 29.20 | 1.45 | 31.46 | 32.13 | 1.99 | 28.33 | 28.95 | 1.70 | 28.46 | 29.08 | 1.69 |
| September | 21.29 | 21.12 | 5.97 | 21.65 | 21.46 | 3.56 | 19.82 | 19.89 | 3.61 | 19.77 | 19.79 | 3.43 |
| October | 14.55 | 15.47 | 6.83 | 15.30 | 17.08 | 6.48 | 15.34 | 16.87 | 6.50 | 15.10 | 16.61 | 6.46 |
| November | 12.58 | 10.32 | 10.79 | 12.76 | 10.12 | 10.96 | 12.83 | 10.49 | 10.39 | 12.66 | 10.36 | 10.29 |
| December | 9.16 | 8.56 | 6.46 | 9.51 | 8.50 | 6.82 | 9.54 | 9.11 | 6.60 | 9.38 | 8.90 | 6.55 |

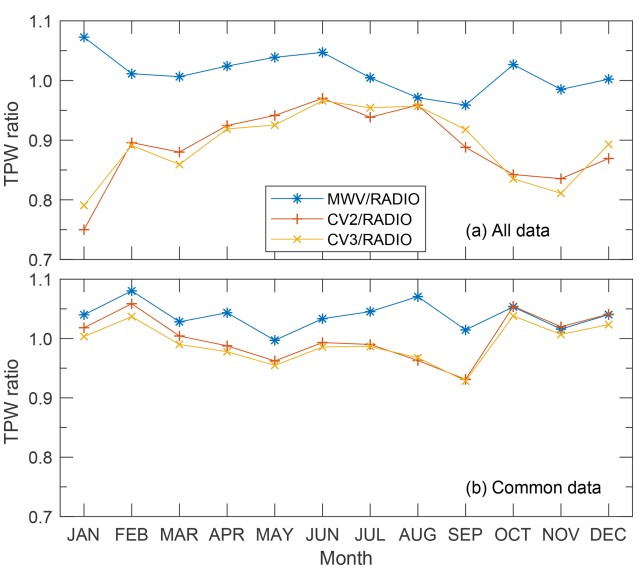

**Figure 4.** Monthly ratio of TPW among microwave radiometer, Cimel V2 and V3, and radiosondes **(a)** for all the available measurements and **(b)** for their datasets.

of factors that could affect the measurements (i.e. the total amount of TPW, the SZA, the sensor temperature and the differences at the AOD at 870 nm) were examined. No significant dependence was found with SZA when comparing the two versions. The relative difference between V2 and V3 show a dependence on TPW (Fig. 10a). The biggest differences (i.e. ∼ 2.5 %) are observed for TPW values lower than 10 mm, while the agreement between the two datasets improves with increased TPW values. However, the decrease in the relative difference of TPW between V2 and V3 is due to the different treatment of the temperature correction in the versions. As shown in Fig. 3 the lowest TPW values appear during wintertime, when the temperature is low as well. Corresponding to these low temperature values the differences between V2 and V3 shows a mean maximum value of ∼ 2.5 % (Fig. 10b). A very pronounced dependence is also seen by the temperature of the internal sensor of the instrument. This dependence is due to the different temperature coefficients in the two versions of the retrieval algorithm. For V2 the temperature coefficients are based on the manufacturer specifications, while in V3 the temperature characterisation is based on laboratory measurements during the calibration of the instrument. The highest positive differences, on the order of ∼ 5 %, appear for low temperatures (< 10 °C). For the whole range of temperatures that are recorded in the instrument (i.e. ∼ 50 °C) a total difference of up to 5 % is observed (Fig. 10b).

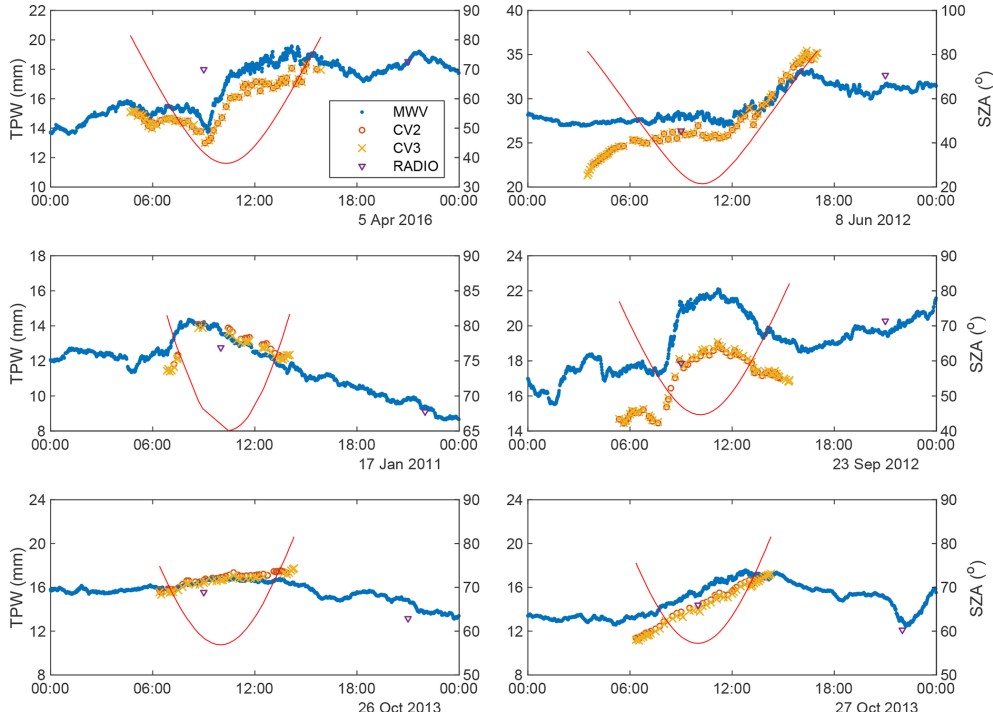

**Figure 5.** Diurnal variation in total precipitable water from the radiosonde (magenta triangle), microwave radiometer (blue dots) and Cimel sun photometer (V2 and V3 of the algorithm, red circle and orange cross, respectively) for 6 selected days (i.e. to cover all seasons and have a relative high number of Cimel sun photometer measurements). The time is in UTC (i.e. local time – 2 h). The red line indicates the range of the SZAs under which Cimel sun photometer measurements were performed.

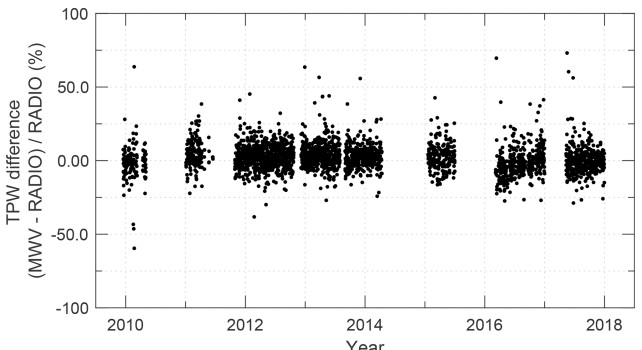

**Figure 6.** Time series of the relative difference (%) between the TPW from the microwave radiometer and the radiosonde during the period 2009–2017.

## 3.5 Comparison between the Cimel sun photometer and radiosondes

To have a better overview about how the differences between the two versions affect the agreement with the other instruments, the evaluation of Cimel sun photometer measurements with radiosondes and the microwave radiometer was based on the common dataset between the two different algorithm versions (Sect. 3.4). Since this Cimel model is not capable of night-time measurements, the comparison

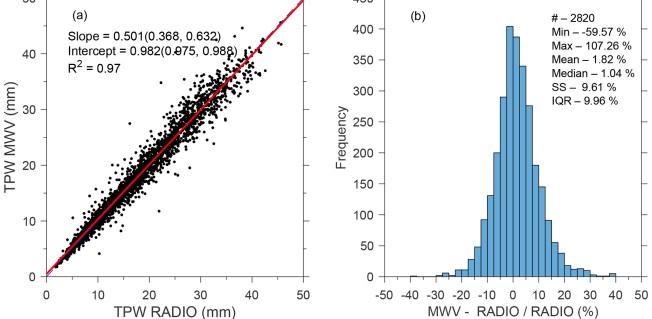

**Figure 7. (a)** Scatter plot of TPW values derived from microwave radiometer and radiosondes. The blue dashed line represents the identity line and the red solid line is the least-square linear fit. The regression coefficients are displayed along with their 95 % confidence interval (in parentheses). **(b)** Frequency distribution of the relative mean difference in TPW between microwave radiometer and radiosondes in bins of 2.5 %.

is limited to daytime measurements only (i.e. radiosondes launched at 12:00 UTC). To account for spatial and temporal differences, the same procedure with the one described for the comparison between microwave radiometer and radiosondes was used (i.e. averaging all Cimel sun photometer points over an interval of 40 min centred on the radiosonde

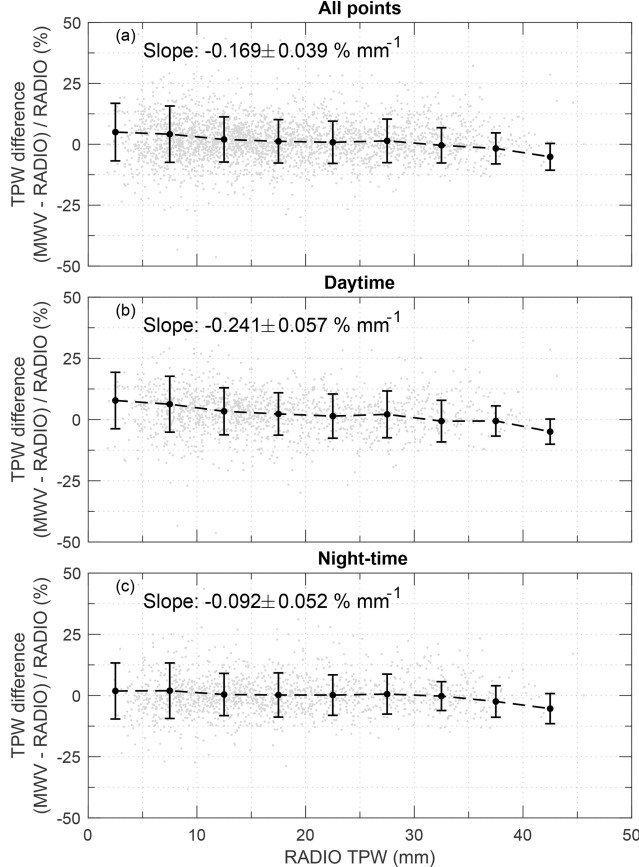

**Figure 8.** Dependence plot of the relative difference of the TPW from the microwave radiometer and the radiosondes from the total amount of TPW for **(a)** all points, **(b)** the daytime measurements and **(c)** the night-time measurements. The black dots show the average difference in bins of 5 mm and the error bars represent their standard deviation. The linear fit is based on all measurements.

launching time). Thus, a total of 682 common measurements were identified.

The differences between the Cimel sun photometer and radiosondes range within $\pm 20\%$ (Fig. 11), while the overall mean difference is $-1.95 \pm 10.97\%$ (or $-0.39 \pm 2.1$ mm) and $-2.74 \pm 10.56\%$ (or $-0.50 \pm 2.05$ mm), for V2 and V3. These results are in agreement with previous studies that showed that AERONET sun photometers generally underestimate TPW in comparison with other instruments (Schneider et al., 2010; Campmany et al., 2010; Pérez-Ramírez et al., 2014; Gui et al., 2017; Campanelli et al., 2018). Version 3 shows an increased underestimation of TPW in comparison with the radiosondes; however the standard deviation is slightly better than in the previous version (Fig. 11b).

The TPW from both versions is highly correlated with the TPW from the radiosondes (i.e. $R^2$ is 0.95 for both Cimel V2 and Cimel V3; Fig. 12a and c), with the slope of the least-square regression line being very close to unity. The histogram of the relative differences between the two datasets

has a very small kurtosis towards negative values, for both Cimel V2 and Cimel V3 (Fig. 12b and d). According to the Shapiro–Wilk test for normality (Shapiro and Wilk, 1965), it does not follow a Gaussian distribution ($p$ value $= 0.01427$ and 0.004603, for V2 and V3, respectively). For Cimel V2 about 65 % of the differences are within $\pm 10\%$, while $\sim$ 93 % are within $\pm 20\%$. For Cimel V3 the respective numbers are 67 % and 93 %. The low number of the coincidence measurements, and their big scatter among different SZAs, TPWs and temperatures of the sensor, does not allow a further evaluation of the influences from these factors.

## 3.6 Comparison between the Cimel sun photometer and radiometer

The comparison between the Cimel sun photometer and the microwave radiometer is based on their coincident measurements, with the microwave radiometer observations averaged over a 1 min interval. This common dataset consists of 8505 observations for the period December 2009–May 2016. The differences between the TPW from both versions of AERONET algorithms are in general within $\pm 10\%$ (Fig. 13). The Cimel sun photometer underestimates the TPW by $2.75 \pm 5.85\%$ (or $0.70 \pm 1.22$ mm) and $3.57 \pm 5.54\%$ (or $0.81 \pm 1.17$ mm), for V2 and V3, respectively. The comparison of the Cimel sun photometer with the MWV reveals a lower overall uncertainty of $\sim 6\%$ estimated as the one sigma of the mean difference, compared to the one ($\sim 10\%$) that was calculated from the comparison of the Cimel sun photometer with the radiosondes. This lower uncertainty can be attributed to the collocation of the Cimel sun photometer and MWV and subsequently the sounding of the same air masses from both instruments. The distance between the RADO site and the radiosonde launching site increases the estimated uncertainty of the retrieved TPW from Cimel sun photometer; however it still remains within the limits that have been estimated by other studies in the past (e.g. Schneider et al., 2010; Pérez-Ramírez et al., 2014).

The TPW values from the Cimel sun photometer (both Cimel V2 and Cimel V3) and the microwave radiometer are highly correlated (Fig. 14a and c; $R^2 = 0.99$). Taking into consideration that the microwave radiometer and the Cimel sun photometer have the same diurnal variations (Sect. 3.2), a very high correlation of the two datasets was expected. For higher values of TPW there is a deviation from the identity line.

The histogram of the relative differences between the two datasets has a very small flattening towards negative values, for both Cimel V2 and Cimel V3 (Fig. 14b and d). According to the Shapiro–Wilk test for normality (Shapiro and Wilk, 1965) ($p$ value $< 2.2$e–16 for both Cimel V2 and Cimel V3) it does not follow a Gaussian distribution. For Cimel V2 about 88 % of the differences lie within $\pm 10\%$, while differences for almost the entire dataset are within $\pm 20\%$ ($> 99\%$). For Cimel V3 the respective values are similar.

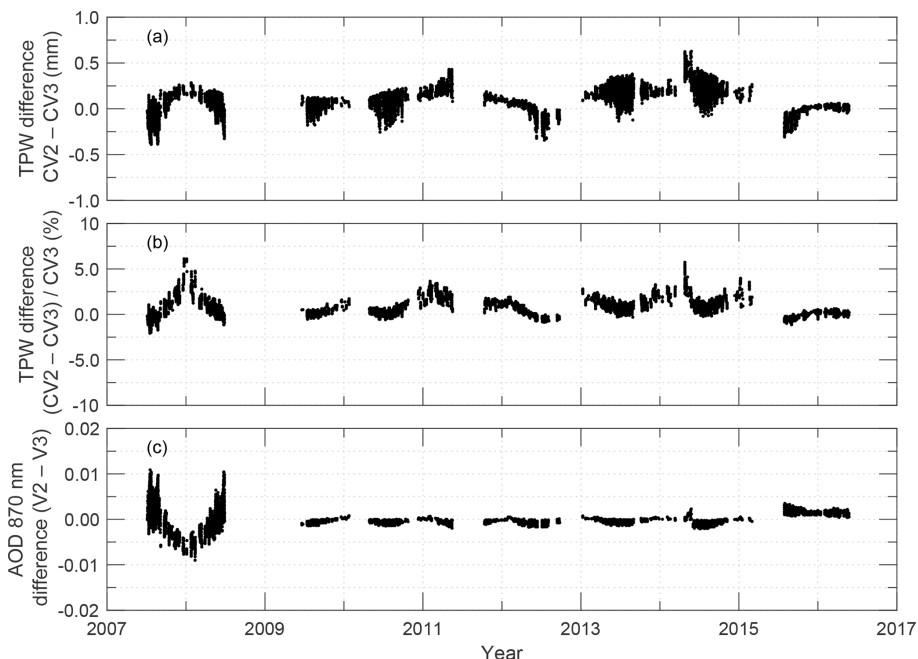

**Figure 9.** Time series of the **(a)** absolute and **(b)** relative differences between level 2.0 of V2 and V3 TPW and **(c)** differences of AOD at 870 nm between V2 and V3 from Cimel sun photometer measurements, for their common measurements during the period 2007–2016.

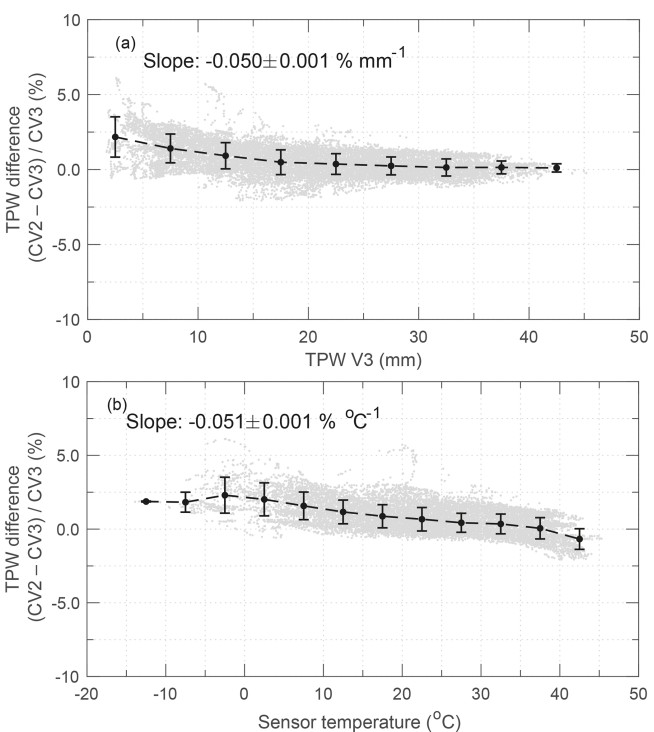

**Figure 10.** Dependence plot of the relative difference of the TPW from V2 and V3 AERONET algorithms from **(a)** the total amount of TPW and from **(b)** the temperature of the censor. The black dots show the average difference in bins of 5 mm and 5 °C, and the error bar represents the standard deviation of the mean. The linear fit is based on all measurements.

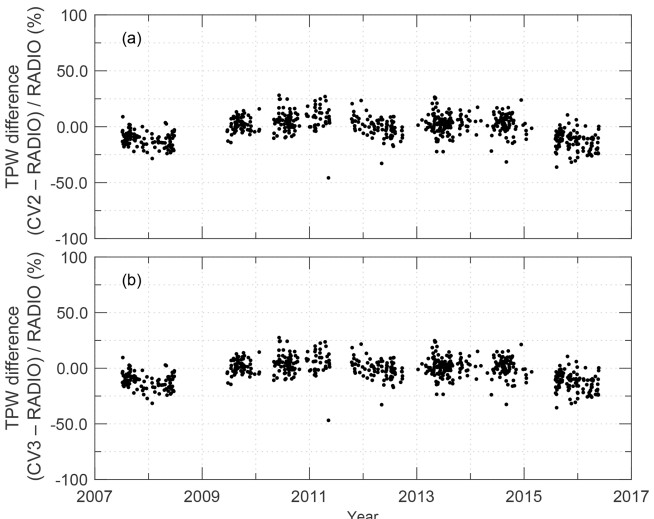

**Figure 11.** Time series of relative differences between **(a)** level 2.0 V2 TPW from the Cimel sun photometer and the radiosondes and **(b)** from level 2.0 V3 TPW from the Cimel sun photometer, during the period 2007–2017.

These results show a very good agreement between the two different methods for the retrieval of TPW.

The difference of the TPW between the Cimel sun photometer and the microwave radiometer does not show a pronounced dependence on the SZA (Fig. 15a and b), for both versions of AERONET algorithms. However, there is an increased scatter for SZAs higher than 70°. This is due to the

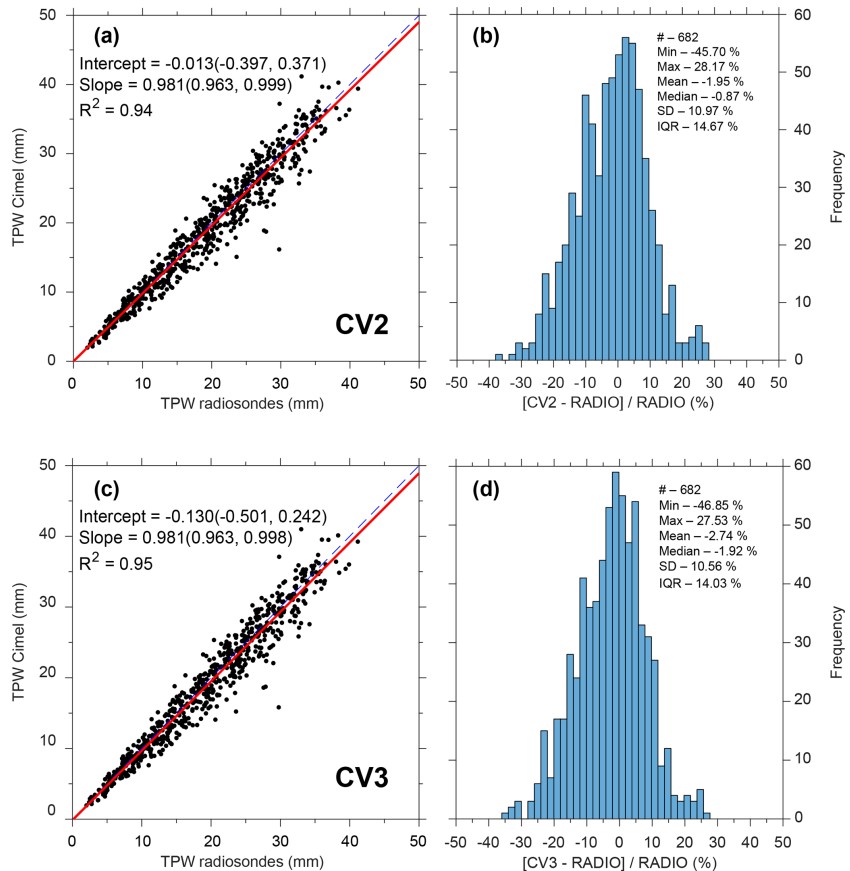

**Figure 12.** Scatter plot between the TPW from **(a)** the radiosondes and Cimel V2 and **(c)** Cimel V3. The red thick line shows the least-square regression line and the blue dashed line is the identity line. The regression coefficients are displayed along with their 95 % confidence interval (in parentheses). Frequency histogram of the relative difference between **(b)** the TPW from the radiosondes and Cimel V2 and **(d)** Cimel V3.

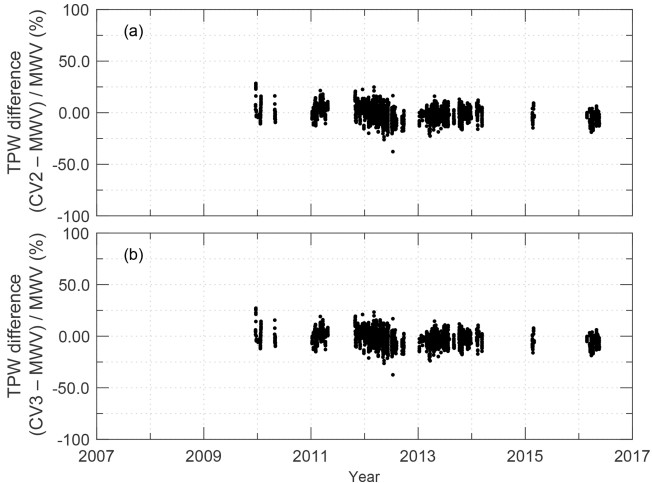

**Figure 13.** Time series of the relative differences between **(a)** level 2.0 V2 TPW from Cimel sun photometer and the microwave radiometer, and from **(b)** level 2.0 V3 TPW from the Cimel sun photometer, during the period 2009–2017.

clock shift effect (see Sect. 3.2) that can affect the direct sun measurements from the Cimel sun photometer at high air masses, resulting in an increased uncertainty on the retrieved TPW.

The difference of the TPW between the Cimel sun photometer and MWV radiometer has a small dependence on the total amount of TPW of $-1.97\%$ per 10 mm for Cimel V2 and $-1.38\%$ per 10 mm for Cimel V3 (Fig. 15c and d). The lower dependence of TPW from Cimel V3 on the total amount of TPW in comparison with Cimel V2 is an indication that the changes applied in the newer version of the algorithm are more correct. Both versions show a higher variability for TPW values lower than 10 mm due to the increased uncertainty of both instruments for dry conditions. However, this variability is based on a relatively low number of observations and is highly affected by some outliers (i.e. differences > 20 %) observed for extremely low TPW values (i.e. 1.5–2 mm). When the TPW values lower than 10 mm are excluded from the analysis, the dependence of the difference between the Cimel sun photometer and MWV radiometer becomes $-1.69\%$ per 10 mm and $-1.19\%$ per 10 mm, for V2 and V3, respectively. In addition the very low variability for

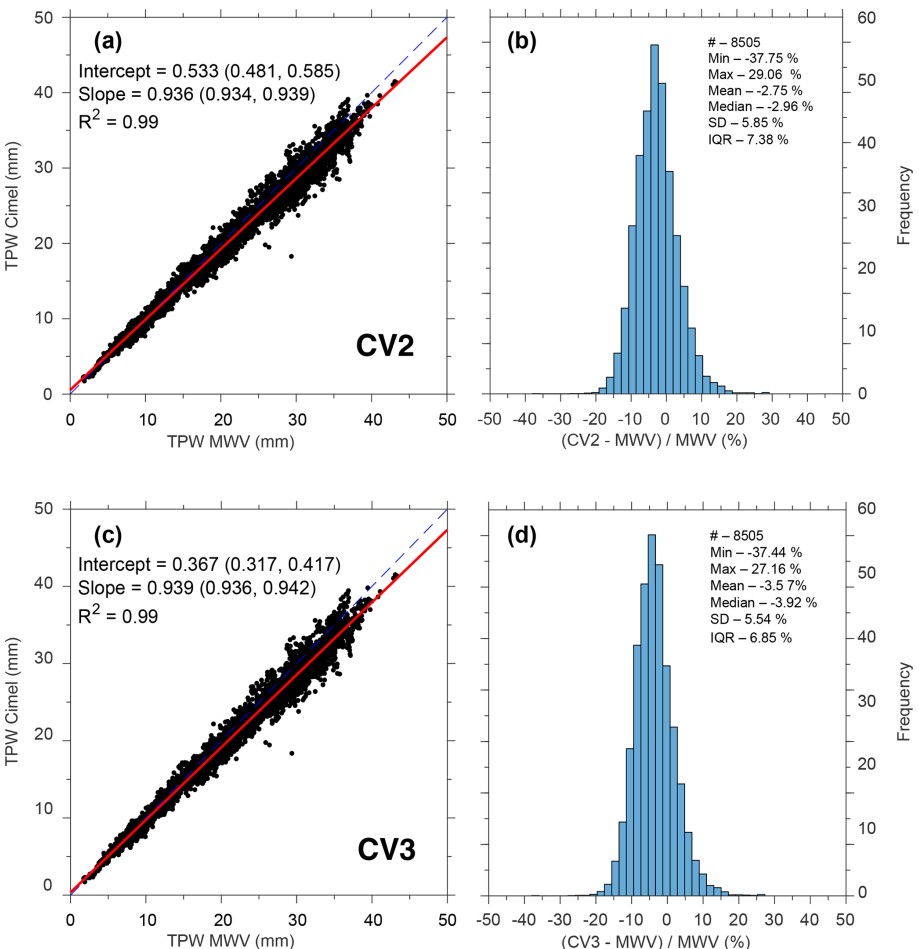

**Figure 14.** Scatter plot between the TPW from **(a)** the microwave radiometer and Cimel V2 and **(c)** Cimel V3. The red thick line shows the least-square regression line and the blue dashed line is the identity line. The regression coefficients are displayed along with their 95 % confidence interval (in parentheses). Frequency histogram of the relative difference between **(b)** the TPW from the microwave radiometer and Cimel V2 and **(d)** Cimel V3, respectively.

TPW values higher than 40 mm cannot be evaluated because they are based on a very limited number of observations (i.e. six observations).

The new laboratory-based temperature coefficients for the sun photometer filters improve the quality of the retrieved TPW from the Cimel sun photometer, as can be depicted from the comparison with the MWV (Fig. 15f). The dependence of the difference between Cimel V3 and the MWV from the temperature recorded in the sensor of the Cimel sun photometer is substantially improved in comparison with the one of Cimel V2 (the order of $-0.61$ % per $10\,°C$ and $-1.07$ % per $10\,°C$ for Cimel V3 and Cimel V2, respectively; Fig. 15e and f). Thus the corrections from the application of the new temperature coefficients are important, since they significantly improve the quality of the retrieved TPW for all the operating temperatures.

## 4   Conclusions

In this study different measurement techniques for TPW (e.g. radiosonde, microwave radiometer, Cimel sun photometer) were compared over a period of 9 years. The microwave radiometer and Cimel sun photometer operated at the RADO situated at a distance of approximately $10\,km$ from the Bucharest city centre. The radiosonde measurements were provided by the Romanian National Meteorological Administration, approximately $30\,km$ from the RADO facilities. The main conclusions of this study can be summarised as follows.

  – All three instruments depict the same annual cycle of TPW despite their different sampling rates. Some small differences observed in the monthly mean values can be attributed to the different schedule (i.e. the microwave radiometer operates during both daytime and nighttime, while the Cimel sun photometer operates only

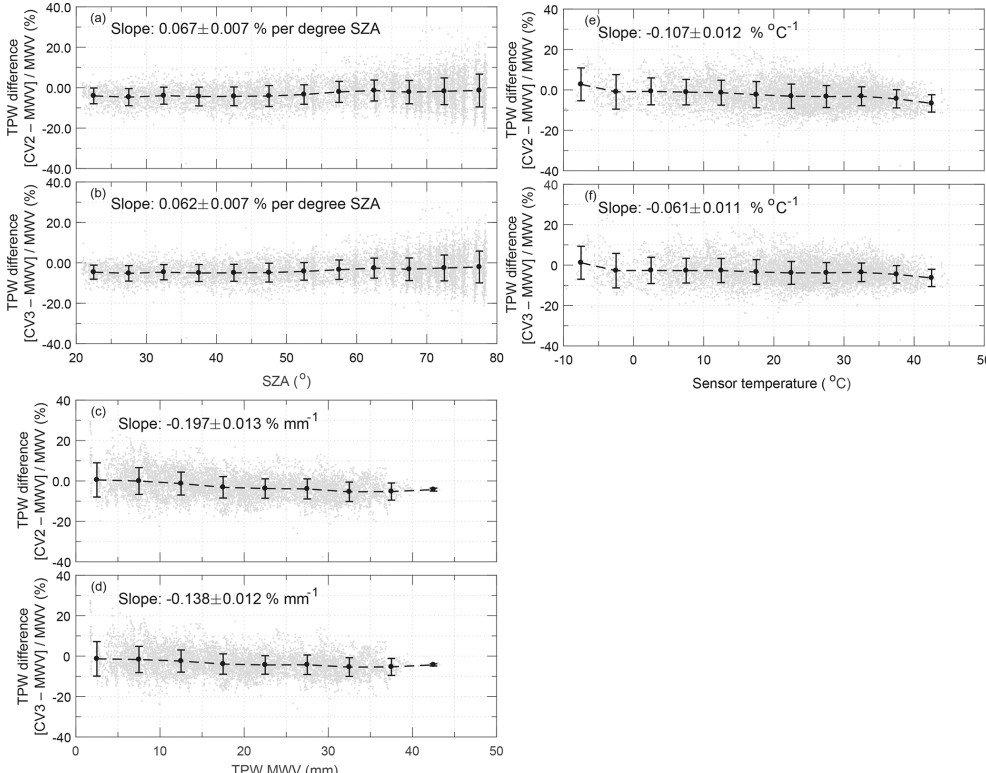

**Figure 15.** Dependence plot of the relative difference of the TPW from Cimel sun photometer and the radiometer from the SZA **(a)** for Cimel V2 and **(b)** Cimel V3. The relative difference between **(c)** Cimel V2 and **(d)** Cimel V3 as a function of TPW and the internal sensor temperature for **(e)** V2 and **(f)** V3. The black dots show the average difference in bins of 5°, 5 mm and 5 °C and the error bar represents the standard deviation of the mean.

during daytime and under clear-sky conditions) and their different sample, partly due to the existing gaps in MWV and Cimel sun photometer.

– The Cimel sun photometer measurements are affected by the clear-sky bias, which is more pronounced during winter and can lead to values lower by up to 25 % for January compared to the radiosondes. The clear-sky bias is almost negligible during summer months.

– The measurements of the microwave radiometer are highly correlated with those from radiosondes (i.e. $R = 0.98$), indicating that the microwave radiometer can capture the environmental changes that lead to variations in TPW.

– Compared with the radiosondes, the microwave radiometer slightly overestimates the TPW, especially during daytime measurements (i.e. $3.12 \pm 9.93$ % or $0.35 \pm 1.71$ mm), due to the dry bias effect, while the difference between the two datasets during night-time is almost negligible (i.e. $0.50 \pm 9.10$ % or $0.001 \pm 1.57$ mm). In addition, the differences between the two datasets during night-time show a very small dependence (i.e. $-0.092 \pm 0.052$ mm$^{-1}$) on the total TPW amount, in

conjunction with the daytime differences that have an increased dependency (i.e. $-0.169 \pm 0.057$ mm$^{-1}$).

– Version 3 of the AERONET algorithm slightly underestimates TPW with an overall difference of $0.60 \pm 20.91$ % ($0.08 \pm 20.14$ mm), compared to version 2.

– The differences of the TPW between versions 2 and 3 AERONET algorithms for their individual common measurements are small (i.e. $\pm 2$ %). The highest differences are observed for low temperatures of the internal sensor (i.e. $< 10$ °C), while the use of new laboratory-based temperature coefficients has an effect of up to 5 % for the whole range of the temperatures recorded by the instrument ($\sim 50$ °C).

– The V2 and V3 AOD 870 nm common values agree within 0.01 AOD and rare larger deviations are likely associated with different temperature coefficients applied in V2 and V3.

– TPW from the Cimel sun photometer is highly correlated with the radiosonde measurements (i.e. $R^2 = 0.99$) for both versions of the AERONET algorithm.

- Compared with the radiosondes, the Cimel sun photometer underestimates the TPW by $1.95 \pm 10.97\%$ (or $0.39 \pm 2.10$ mm) for V2 and $2.74 \pm 10.54\%$ (or $0.50 \pm 2.05$ mm) for V3. This underestimation is in agreement with previous studies comparing measurements from radiosondes and sun photometers for different regions.

- When compared with the microwave radiometer, the Cimel sun photometer underestimates by $2.75 \pm 5.85\%$ (or $0.70 \pm 1.22$ mm) for V2 and $3.57 \pm 5.54\%$ (or $0.81 \pm 1.17$ mm) for V3. The two instruments have the same daily cycle, which shows the capability of the Cimel sun photometer to capture the daily variations in TPW. However, some discrepancies are observed during early morning or late afternoon, which are induced from a shift in the Cimel clock resulting in a minor error in the calculation of the optical air mass. However, changes in the Cimel TPW are within uncertainty estimates. While the difference between the Cimel sun photometer and radiometer does not show any pronounced dependence on SZA, for SZAs $> 70°$ the differences show an increased scatter.

- V3 has a lower dependence from the total TPW amount (i.e. $-0.138 \pm 0.012$ mm$^{-1}$) compared with V2 (i.e. $-0.197 \pm 0.013$ mm$^{-1}$). The new laboratory-based temperature coefficients implemented in V3 reduced the dependence of the recorded differences between the Cimel sun photometer and the microwave radiometer (i.e. $-0.107 \pm 0.012$ °C$^{-1}$ and $-0.061 \pm 0.011$ °C$^{-1}$ for V2 and V3, respectively).

- The implementation of the new temperature coefficients in V3 has significantly improved the quality of the retrieved TPW and AOD from Cimel sun photometer measurements, especially before 2009, when the filter at 870 nm had higher sensitivity to temperature variations.

To our knowledge this is the first study to evaluate, in depth, the TPW retrieval from the newly released version 3 of the AERONET algorithm. The comparison with high-quality independent measurements from radiosondes and a collocated radiometer shows that the absolute level of the differences in V3 from the other instruments is a little higher than in V2. However, the one-sigma uncertainty for V3 compared to the radiosondes is $\sim 10\%$, which is in accordance with previous studies for V2. This slightly increased uncertainty could be attributed to the relatively high distance between the Cimel sun photometer and the radiosonde launching site. Compared with the collocated MWV radiometer the estimated uncertainty is further reduced to less than 6%. V3 has a lower dependence on the TPW and the internal sensor temperature, which in principle should improve the TPW Cimel retrievals. Nevertheless, further evaluation is needed, especially for sites with different characteristics (i.e. mountain or marine environments). Although these findings are for

a specific site, they are likely representative for other continental sites as well. A future study will investigate the accuracy of the night-time TPW from Cimel lunar measurements, available at the RADO facilities since 2016, following the methodology applied in this study. Finally, the microwave radiometer shows a very good performance compared with the radiosondes, especially during night-time when the differences between the two instruments are almost negligible. Thus, the microwave radiometer can be used in future studies related to the validation of satellite datasets during both daytime and night-time.

*Data availability.* The data from the radiosondes for Bucharest (station ID: 15420) are publicly available through the upper air observations database of the University of Wyoming at the link http://weather.uwyo.edu/upperair/sounding.html (last access: 13 July 2018). The Cimel sun photometer data can be found at the AERONET website (https://aeronet.gsfc.nasa.gov/, last access: 14 January 2019) under the label Bucharest_Inoe. The data from the microwave radiometer and the relative humidity from the meteorological station are available upon request. The sunshine duration and cloud faction are available through the EUMETSAT's CM SAF web portal (https://wui.cmsaf.eu/safira/action/viewProduktSearch, last access: 29 November 2018).

*Author contributions.* KF and GAE initiated the idea for this paper. KF performed the analysis for the biggest part of the paper with the assistance of BA. DE was responsible for the calibration of the microwave radiometer and produced the L2 data from the instrument. DMG provided important information about the V3 AERONET and reviewed parts of the comparison of the two algorithms and of Cimel sun photometer with the other instruments. MB performed the analysis for the sunshine duration and cloud coverage. LV was responsible in the past for the calibration of the microwave radiometer and local site manager of the Cimel sun photometer. DN is the PI of the Bucharest_INOE AERONET station. KF and BA prepared the paper with contributions from all co-authors.

*Competing interests.* The authors declare that they have no conflict of interest.

*Acknowledgements.* The authors would like to thank the two anonymous reviewers for their constructive comments that significantly improved the paper. We thank AERONET (PHOTONS) for instrument calibration and maintenance of the Cimel instrument and AERONET (GSFC) for processing and disseminating these data. The authors would further like to acknowledge the EUMETSAT CM-SAF project team for providing the climate variables used in this study. This work has received funding from the European Union's Horizon 2020 Research and Innovation Programme, under grant agreement no. 692014, project ECARS (East European Centre for Atmospheric Remote Sensing). Part of the work performed for this study was funded by the Ministry of Research and Innovation through Program I – Development of the national

research–development system, Subprogram 1.2 – Institutional Performance – Projects of Excellence Financing in RDI, contract no. 19PFE/17.10.2018 and by Romanian National Core Program contract no. 18N/2019.

*Review statement.* This paper was edited by Laura Bianco and reviewed by Ioannis Panagiotis Raptis and one anonymous referee.

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

**Remarks from the typesetter**

TS1    Please provide last access date and please confirm the reference.