# Peer review of "Assessment of the total precipitable water from a sun-photometer, microwave radiometer and radiosondes at a continental site in southeastern Europe"

_Atmospheric Measurement Techniques, 2018_

## Referee Comment (RC1) · Anonymous Referee #2 · 29 Oct 2018

This study is very interesting, fitting the scope of ATM journal and could prove very useful for future studies in the field. Intercomparison of TPW retrieved from different techniques, for decade long timeseries is not found regularly in the literature and at this work data from radiosondes, sunphotometer and microwave radiometer are compared at such interval. Authors aim more to compare two versions of AERONET retrieving algorithm, which is very interesting but the results found are not discussed in depth, differences and uncertainty is not justified in depth. Thus, I suggest to consider the manuscript for publication after undertaking major revision.

 More specific com-

ments:

a) Various retrieving approaches have been studies in other publications, but authors should make clear which approach is considered more representative of actual atmospheric conditions. To my knowledge, radiosonde retrievals, being real in situ measurements at different heights, are the data that should have this role. Although there are uncertainty at this retrieval. In my opinion all comparisons and explanation should be a performed according to the principle that more reliable are remote sensing retrievals closest to radiosondes. b) To my knowledge, there is no other publication for the version 3 algorithm of AERONET. Thus, a more detailed discussion on this algorithm is needed. Especially at paragraph 2.1 , formulas and hypothesis used for retrieving TWP should be discussed. A very important aspect, is that algorithm uses AOD retrievals from other wavelengths, hence the differences to AOD between the two versions, should be propagated also to TWP. Also, some discussion is needed on the uncertainty of this retrieval and if there are any differences to it between the two versions. c) I have very serious doubt on the data from radiosondes and microwave radiometer. Sunphotometric retrievals usually are up to around 40 mm, because higher loads are usually linked to the presence of clouds. Hence, radiosondes and microwave radiometer that are measuring no matter the cloud conditions should have significant higher average and median values (table 1, figure 1). Either authors have used only synchronous to aeronet data for all retrievals, thus cloud conditions are filtered out or there is some severe problem with the datasets. This should be clarified in any of those cases and rewrite this paragraphs to clarify the procedure or recalculate all statistics. At p5 l4 , it is stated that for daily mean values all measurements are used. d) More information on the climatology of the measuring site is needed at paragraph 2. I would suggest at least some statistics on yearly sunshine hours (which affects the quantity of aeronet data) and some range and averages for water vapor or at least moisture in the area. e) Since the normal distribution is visualized on histograms (red line - figures 5,10,12), it is expected to perform some statistical test to determine whether data's distribution fits to it. I suspect from the plots that it doesn't fit , so it is preferable to find the distri-

bution that best fits the data and at least some discussion in the manuscript should be expected in the manuscript. f) At paragraph 2.3 some discussion on the spatial spread of radiosondes should be added. I would suggest to filter out some radiosondes from the comparisons, with some criterion about the final position or at least the position at 4km height, which could make data uncomparable, since the distance between the two sites is already 30km and in case of southwest winds could be even larger for the sounding. g) Paragraph 3.2 needs clarification. What "randomly selected days" mean? How the random process was performed? What does "enough cimel observations" mean? Please be more clear when describing these procedures and make clear what conditions were applied and how was selected. h) P6 l10. This sentence should be more clear and have a more extensive discussion. What sza corresponds to these airmasses? Is that the airmasses that differs between v2 and v3? Higher sza values at 8 june and 26 october should be a lot different, are these differences observed at the same sza's at these days? Is that pattern observed at other days at these angles? I would suggest to add a plot in figure 3 with sza at x-axis to make all this clearer. i) p6 l13. I would suggest to use the more robust approach found at Schneider et al. (2010) averaging measurements for $\pm$ 20 min from the time that the radiosonde reaches a 4 km height, in order to minimize spatial and temporal measurement differences.
 j) paragraph 3.4. I strongly suggest to investigate the differences in respect to AOD at 870 nm . Following the earlier about the missing methodology for TWP retrieval, I suspect that measurements with high differences in AOD between v2 and v3 will propagate to TWP values, especially at values less than 10mm, where the AOD influence is a lot larger. k) P8 l5, so it is suggested that v3 has slightly less accordance to the more reliable measurement. I suggest to use the Schneider criterion for averaging data around radiosonde, to have a more robust estimation and also I think that AOD differences will partially explain this behavior. Otherwise, it would be an interesting finding that v3 downgraded the quality of TWP. l) P8 l32. This trend is calculated statistically, but for lower values there is a very high spread of differences, thus I have doubt if this statistics is meaningful. Practically values below 10mm could have any difference, and

higher values converge a lot. This behavior is explained through the uncertainty of both instruments that lowers for higher TWP. This should be discussed in respect to uncertainty estimations. m) P19 l15. What is the distance to radiosonde launching site? 18 or 30 km (stated in paragraph 2.3)

Schneider, M., Romero, P. M., Hase, F., Blumenstock, T., Cuevas, E., & Ramos, R., Continuous quality assessment of atmospheric water vapour measurement techniques: FTIR, Cimel, MFRSR, GPS, and Vaisala RS92. Atmospheric Measurement Techniques, 3(2), 323–338. http://doi.org/10.5194/amt-3-323-2010,2010.

---

## Referee Comment (RC2) · Anonymous Referee #1 · 31 Oct 2018

This work is a comprehensive comparison of different measurements (Cimel sun-photomer versions 2 and 3, microwave radiometer and radiosondes) of TPW for a single station close to Budapest. It is generally well written, however I strongly recommend the help of a native speaker to improve the language.

The article has improved from initial submission, but still I would like to point out some issues that should be address.

1.  First of all, the methodology section is still missing, although the authors claimed

that they were including it. Maybe there was some problem with the manuscript resubmission. This methodology section should say how are the statistics computed, as well as the matching criteria.

2. Section 2: More information on how the IWV is retrieved from the instrument measurements would be desirable.

3. Page 2, L.26: Actually, as far as I know, sun-photometers only need that the solar disc is free from clouds, but the rest of the sky can be covered by clouds. So the phrase is not really correct.

4. Page 3, L. 6: Since the models of the radiometer and the sun-photomer is mentioned, the radisondes model should also appear in this line.

5. Page 4, L. 17-18: authors should explain this "visually inspection". Was it looking at the time-series plot? By comparison with the other instruments?

6. Page 6, L. 23: could the authors provide a reference to the dry bias effect?

7. Page 6, L. 30-31. Why is the dependence almost negligible? Could the authors explain this fact?

8. Page 7, L. 2: could the authors specify the number of data?

9. Page 7, L. 30-31: could the authors provide some reference to this issue?

10. Page 8, L. 28: I do not agree with this sentence, since the Version 3 is supposed to be better quality than Version 2, so the phrase does not make much sense.

11. Scatterplot figures (Fig 5a, 10, 12, etc.): authors could provide the confidence interval for the coefficients of regression. Also, p-values cannot be exactly equal to 0, please use the scientific notation to indicate the order of magnitude. Also, indicate if "corr" refers to $R$ or $R^2$.

12. Histogram of relative differences figures (Fig 10b, 10d, etc): Please, use a statistical test to check if the distribution really follow a normal distribution.

13. Table 2: indicate that these are daily means in the table caption

Technical corrections:

1. Page 1, L.17: "IPCC (2013)" should be "(IPCC, 2013)"

2. Page 2, L. 5: Citation is incorrect.

3. Page 2, L.19: TWP instead of TPW.

4. Page 2, L.22: it should be AErosol RObotoic NETwork (AERONET).

5. Page 5, L.13: I understand PC means Personal Computer, but as an acronym it should be indicated.

6. Page 8, L. 13: maybe the authors mean "do not allow".

---

## Author Comment (AC1) · 30 Jan 2019

We would like to thank the reviewer for his constructive comments that help us to improve the manuscript. Taking into consideration the comments of both reviewers we have made several modifications in the manuscript and the new paper now includes two new sections (section 2.1 and section 2.5), two new figures (figure 1 and 4) and two new tables (table 4 and 5). In addition three more co-authors have been added in the manuscript: David M. Giles, Mihai Boldeanu and Doina Nicolae. David M. Giles provided important information about the V3 AERONET and reviewed parts of the comparison of the two algorithms and of Cimel with the other instruments. Mihai Boldeanu performed the analysis for the sunshine duration and cloud coverage. Doina Nicolae had not been included in the original submission by mistake, since she is the PI of the AERONET station. We have also added the author contribution section in order to justify the necessity for their inclusion in the paper.

In the following with black are the original comments and with blue our replies.

Reviewer #2

This study is very interesting, fitting the scope of ATM journal and could prove very useful for future studies in the field. Intercomparison of TPW retrieved from different techniques, for decade long timeseries is not found regularly in the literature and at this work data from radiosondes, sunphotometer and microwave radiometer are compared at such interval. Authors aim more to compare two versions of AERONET retrieving algorithm, which is very interesting but the results found are not discussed in depth, differences and uncertainty is not justified in depth. Thus, I suggest to consider the manuscript for publication after undertaking major revision.

More specific comments:

a) Various retrieving approaches have been studies in other publications, but authors should make clear which approach is considered more representative of actual atmospheric conditions. To my knowledge, radiosonde retrievals, being real in situ measurements at different heights, are the data that should have this role. Although there are uncertainty at this retrieval. In my opinion all comparisons and explanation should be a performed according to the principle that more reliable are remote sensing retrievals closest to radiosondes.

Answer: In the current version of the manuscript we have followed the reviewer's comment and provided a more detailed description of the comparison between the different instruments (see section 2.5 – Methodology and especially page 7, lines 5-12).

b) To my knowledge, there is no other publication for the version 3 algorithm of AERONET. Thus, a more detailed discussion on this algorithm is needed. Especially at paragraph 2.1, formulas and hypothesis used for retrieving TWP should be discussed. A very important aspect, is that algorithm uses AOD retrievals from other wavelengths, hence the differences to AOD between the two versions, should be propagated also to TWP. Also, some discussion is needed on the uncertainty of this retrieval and if there are any differences to it between the two versions.

Answer: On page 4, lines 23-29 we now provide the following explanation:

"Since Giles et al. (2019) provide a full description of the TPW retrieval algorithm (see Section 2 of that paper), in this section just the major differences between V2 and V3 and some other factors that may influence the TPW retrieval are discussed. For the computation of TPW a necessary preliminary step is the subtraction of the AOD and Rayleigh optical depths from the total optical depth at 935 nm. Since AOD is not calculated direct for the 935 nm channel due to the strong effect of water vapor, the AOD at 870 nm is extrapolated at the 935 nm using the Ångstrom Exponent (AE) at 440-870 nm. The main differences in the computation of TPW in V3 are that the new algorithm accounts for updated continuum look-up table (Mlawer et al., 2012), using Total Internal Partition Sums (Gamache et al., 2017) and using the extraterrestrial spectral solar irradiance from Coddington et al. (2016)."

and in page 5 lines 1-2, "Details about all the improvements implemented in the V3 of AERONET can be found at Giles et al. (2019)."

c) I have very serious doubt on the data from radiosondes and microwave radiometer. Sunphotometric retrievals usually are up to around 40 mm, because higher loads are usually linked to the presence of clouds. Hence, radiosondes and microwave radiometer that are measuring no matter the cloud conditions should have significant higher average and median values (table 1, figure 1). Either authors have used only synchronous to aeronet data for all retrievals, thus cloud conditions are filtered out or there is some severe problem with the datasets. This should be clarified in any of those cases and rewrite this paragraphs to clarify the procedure or recalculate all statistics. At p5 l4 , it is stated that for daily mean values all measurements are used.

Answer: In the methodology section we have now included a more detailed description of how the dataset was constructed (page 6, lines 13-22). Also, to convince the reviewer for the validity of our data two more discussion paragraphs were added in results section (page 7, lines 26 – 31 continuing in page 8, lines 1-7) and page 8 (lines 19-33), which were complemented by two new tables (table 4 and 5) and a new Figure (Fig4). Indeed in our dataset we observed the clear sky bias mentioned by the reviewer, but we only observe this in the monthly averages and not in the overall averages, as explained in the manuscript.

d) More information on the climatology of the measuring site is needed at paragraph 2. I would suggest at least some statistics on yearly sunshine hours (which affects the quantity of aeronet data) and some range and averages for water vapor or at least moisture in the area.

Answer: We have now added a new section (2.1 Meteorological parameters) where we provide statistics concerning relative humidity, sunshine duration and cloud fraction (see also Figure 1).

e) Since the normal distribution is visualized on histograms (red line - figures 5,10,12), it is expected to perform some statistical test to determine whether data's distribution fits to it. I suspect from the plots that it doesn't fit, so it is preferable to find the distribution that best fits the data and at least some discussion in the manuscript should be expected in the manuscript.

Answer: We have used the Shapiro-Wilk test to check the normality of the data ((page 9, lines 25-26, page 11, lines 14-15 and page 11, line 35). We have now deleted the normal distribution fit from figures.

f) At paragraph 2.3 some discussion on the spatial spread of radiosondes should be added. I would suggest to filter out some radiosondes from the comparisons, with some criterion about the final position or at least the position at 4km height, which could make data uncomparable, since the distance between the two sites is already 30km and in case of southwest winds could be even larger for the sounding.

Answer: Because we do not have access to the raw radiosonde data, we are not able to apply the filter suggested by the reviewer. We have included in the methodology section the relative discussion about this issue and how it could affect the estimated uncertainties (page 6, lines 19-25).

g) Paragraph 3.2 needs clarification. What "randomly selected days" mean? How the random process was performed? What does "enough cimel observations" mean? Please be more clear when describing these procedures and make clear what conditions were applied and how was selected.

Answer: The word "random" was a bad choice and have now been deleted. A clarification has been added on page 9 (lines 6-7).

"The days were selected under the condition that the Cimel measurements cover the biggest part of the day, and especially the high SZAs (>70$^\circ$) no discontinuation due to clouds in Cimel measurements was observed from sunrise to sunset and cover all seasons.

h) P6 l10. This sentence should be more clear and have a more extensive discussion. What sza corresponds to these airmasses? Is that the airmasses that differs between v2 and v3? Higher sza values at 8 june and 26 october should be a lot different, are these differences observed at the same sza's at these days? Is that pattern observed at other days at these angles? I would suggest to add a plot in figure 3 with sza at x-axis to make all this clearer.

Answer: On Figure 4 we have added the SZA that correspond to Cimel V3 data measurements, to show the range of SZAs that the measurements were performed.

i) p6 l13. I would suggest to use the more robust approach found at Schneider et al. (2010) averaging measurements for ± 20 min from the time that the radiosonde reaches a 4 km height, in order to minimize spatial and temporal measurement differences.

Answer: Unfortunately due to the limitation of the radiosonde dataset we are unable to apply the approach from Schneider et al. (2010). See also reply to comment f)

j) paragraph 3.4. I strongly suggest to investigate the differences in respect to AOD at 870 nm. Following the earlier about the missing methodology for TWP retrieval, I suspect that measurements with high differences in AOD between v2 and v3 will propagate to TWP values, especially at values less than 10mm, where the AOD influence is a lot larger.

Answer: To address this comment, we have now included in figure 9 the differences of the AOD at 870 nm between V2 and V3, which are then discussed on page 9, lines 8-15:

"The differences at the AOD at 870 nm between the two different algorithm versions (Fig. 9 c) are generally pretty low and rarely exceed the ±0.01 AOD units. The cyclic nature of the AOD differences (Fig. 9 c) suggests the variation of the AOD with temperature for Version 2. The V2 data are not temperature corrected for the 870nm filter and this produces a difference in AOD between temperature corrected (V3) and not corrected (V2) due to this specific filter before 2009. The 870nm filter was changed in 2009 in this specific instrument and its dependence on temperature was a magnitude lower than the initial filter. As a result, the filter used in the instrument from 2009 and onward shows less deviation from V2 since the temperature correction needed for the filter is minimal. This is a clear example of how implementation of temperature correction in Version 3 significantly improved the AOD and TPW, before 2009."

k) P8 l5, so it is suggested that v3 has slightly less accordance to the more reliable measurement. I suggest to use the Schneider criterion for averaging data around radiosonde, to have a more robust estimation and also I think that AOD differences will partially explain this behavior. Otherwise, it would be an interesting finding that v3 downgraded the quality of TWP.

Answer: Because we cannot apply the Schneider criteria to our dataset, it is very difficult to make any assumptions about the degrading in quality in V3. However, the lower dependence of the new version from the TPW and the internal sensor temperature are indications that the retrieval is more robust.

l) P8 l32. This trend is calculated statistically, but for lower values there is a very high spread of differences, thus I have doubt if this statistics is meaningful. Practically values below 10mm could have any difference, and higher values converge a lot. This behavior is explained through the uncertainty of both instruments that lowers for higher TWP. This should be discussed in respect to uncertainty estimations.

Answer: When we exclude from the calculation of the trend the TPW values lower than 10 mm, we still obtain a lower dependence on TPW for V3. We have added the following sentence in the manuscript (page 12, lines 13-15)

"When the TPW values lower than 10 mm are excluded from the analysis, the dependence of the difference between Cimel and MWR becomes -1.69\% per 10 mm and -1.19\% per 10 mm, for V2 and V3 respectively."

m) P19 l15. What is the distance to radiosonde launching site? 18 or 30 km (stated in paragraph 2.3)

Answer: It is 30 km, we would like to thank the reviewer for pointing this out. It has been corrected now.

[revised manuscript text omitted]
 | 9.16 | 8.56 | 6.46 | 9.51 | 8.50 | 6.82 | 9.54 | 9.11 | 6.60 | 9.38 | 8.90 | 6.55 |

---

## Author Comment (AC2) · 30 Jan 2019

We would like to thank the reviewer for his constructive comments that help us to improve the manuscript. Taking into consideration the comments of both reviewers we have made several modifications in the manuscript and the new paper now includes two new sections (section 2.1 and section 2.5), two new figures (figure 1 and 4) and two new tables (table 4 and 5). In addition three more co-authors have been added in the manuscript: David M. Giles, Mihai Boldeanu and Doina Nicolae. David M. Giles provided important information about the V3 AERONET and reviewed parts of the comparison of the two algorithms and of Cimel with the other instruments. Mihai Boldeanu performed the analysis for the sunshine duration and cloud coverage. Doina Nicolae had not been included in the original submission by mistake, since she is the PI of the AERONET station. We have also added the author contribution section in order to justify the necessity for their inclusion in the paper.

In the following with black are the original comments and with blue our replies.

Reviewer #1

This work is a comprehensive comparison of different measurements (Cimel sunphotomer versions 2 and 3, microwave radiometer and radiosondes) of TPW for a single station close to Budapest. It is generally well written, however I strongly recommend the help of a native speaker to improve the language. The article has improved from initial submission, but still I would like to point out some issues that should be address.

1) First of all, the methodology section is still missing, although the authors claimed that they were including it. Maybe there was some problem with the manuscript resubmission. This methodology section should say how are the statistics computed, as well as the matching criteria.

Answer: We have now added the methodology section (section 2.5)

2) Section 2: More information on how the IWV is retrieved from the instrument measurements would be desirable.

Answer: We have enhanced the discussion on the retrieval methods, by adding new paragraphs in section 2.2 (page 4, lines 23-29) and in section 2.3 (page 5, lines 11-15 and 20-29).

3) Page 2, L.26: Actually, as far as I know, sun-photometers only need that the solar disc is free from clouds, but the rest of the sky can be covered by clouds. So the phrase is not really correct.

Answer: We have rephrased now this sentence (and in whole manuscript) and now it is written: "which indicate that at least the solar disc must be free from clouds for TPW retrieval (page 2, line 29)"

4) Page 3, L. 6: Since the models of the radiometer and the sun-photomer is mentioned, the radisondes model should also appear in this line

Answer: We have added the radiosondes type (Vaisala RS92) (page 3, line 10)

5) Page 4, L. 17-18: authors should explain this "visually inspection". Was it looking at the time-series plot? By comparison with the other instruments?

Answer: We have extended the discussion to explain better what we meant by visually inspection and the possible instrumental malfunctions that were identified (page 5, line 20-29). The retrieved Integrated Water Vapor was examined along with the Liquid Water Path, for identification of abnormal values of IWV. An example is given in the figure below. With blue are the original time-series of the IWV, where only the periods with rain have been removed and with red the final, after removing bad quality data.

[Figure]

A closer look at about 15:00 o'clock reveals that there are data identified as rain (missing points) and the increase in the IWV is associated with an increased in the LWP, as well. Furthermore, there is a period with no changes in the IWV and LWP levels indicating a possible problem in the data submission. For these reasons it was decided the data between 14:50 and 15:15 to be removed from the final dataset. A similar situation happens between 17:05 and 17:30.

[Figure]

6) Page 6, L. 23: could the authors provide a reference to the dry bias effect?

Answer: We have added reference (Vömel et al., 2007)

7) Page 6, L. 30-31. Why is the dependence almost negligible? Could the authors explain this fact?

Answer: This underestimation is more evident during daytime (i.e., 3.12±9.93% or 0.35±1.71 mm) due to the radiation dry bias effect that affect the radiosondes (e.g., Vömel et al., 2007), which is more pronounced for TPW values less than 10 mm. During nighttime the differences are almost negligible (i.e., -0.50±9.10% or -0.01±1.57 mm).

8) Page 7, L. 2: could the authors specify the number of data?

Answer: We have added the number of data: (i.e., just 19 measurements)

9) Page 7, L. 30-31: could the authors provide some reference to this issue?

Answer: In the current version of the manuscript we have included a discussion of this issue in the methodology section (page, lines 18-25).

10) Page 8, L. 28: I do not agree with this sentence, since the Version 3 is supposed to be better quality than Version 2, so the phrase does not make much sense.

Answer: Indeed, V3 is better than V2 but as figure 13a and b shows both versions do not have a pronounced dependence from the SZA.

11) Scatterplot figures (Fig 5a, 10, 12, etc.): authors could provide the confidence interval for the coefficients of regression. Also, p-values cannot be exactly equal to 0, please use the scientific notation to indicate the order of magnitude. Also, indicate if "corr" refers to R or R2 .

Answer: We have updated the figures to include the 95% confidence intervals for the regression coefficients, deleted p-values and added the square of the correlation coefficient.

12) Histogram of relative differences figures (Fig 10b, 10d, etc): Please, use a statistical test to check if the distribution really follow a normal distribution.

Answer: We have used the Shapiro-Wilk test to check the normality of the data (page 9, lines 25-26, page 11, lines 14-15 and page 11, line 35).

13) Table 2: indicate that these are daily means in the table caption

Answer: We have now updated the table caption

Technical corrections:

1. Page 1, L.17: "IPCC (2013)" should be "(IPCC, 2013)"

Answer: Done

2. Page 2, L. 5: Citation is incorrect.

Answer: We have corrected it

3. Page 2, L.19: TWP instead of TPW.

Answer: Corrected

4. Page 2, L.22: it should be AErosol RObotoic NETwork (AERONET).

Answer: Done

5. Page 5, L.13: I understand PC means Personal Computer, but as an acronym it should be indicated.

Answer: Done

6. Page 8, L. 13: maybe the authors mean "do not allow"

Answer: The reviewer is right; we would like to thank him for pointing this out

[revised manuscript text omitted]
 | 9.16 | 8.56 | 6.46 | 9.51 | 8.50 | 6.82 | 9.54 | 9.11 | 6.60 | 9.38 | 8.90 | 6.55 |

---

## Referee Report (RR1)

The revised version of the manuscript answers all the reviewers concerns and some paragraphs have been rewritten/restated to clarify the approach followed by authors. All comments have been addressed and explained/corrected.

Thus I recommend the manuscript to be accepted for publications, since the study and evaluation of water vapor from AERONET version 3 algorithm is of great significance for various fields of atmospheric science.